# Neural representational geometries reflect behavioral differences in monkeys and recurrent neural networks

Valeria Fascianelli [1,2] ✉, Aldo Battista [3], Fabio Stefanini[1,2], Satoshi Tsujimoto[4], Aldo Genovesio [5] ✉ & Stefano Fusi [1,2,6,7] ✉

Animals likely use a variety of strategies to solve laboratory tasks. Traditionally, combined analysis of behavioral and neural recording data across subjects employing different strategies may obscure important signals and give confusing results. Hence, it is essential to develop techniques that can infer strategy at the single-subject level. We analyzed an experiment in which two male monkeys performed a visually cued rule-based task. The analysis of their performance shows no indication that they used a different strategy. However, when we examined the geometry of stimulus representations in the state space of the neural activities recorded in dorsolateral prefrontal cortex, we found striking differences between the two monkeys. Our purely neural results induced us to reanalyze the behavior. The new analysis showed that the differences in representational geometry are associated with differences in the reaction times, revealing behavioral differences we were unaware of. All these analyses suggest that the monkeys are using different strategies. Finally, using recurrent neural network models trained to perform the same task, we show that these strategies correlate with the amount of training, suggesting a possible explanation for the observed neural and behavioral differences.

Although the tasks designed in a laboratory are relatively simple and are performed in highly controlled situations, different animals can still adopt different strategies to solve the same task. It is surprisingly difficult to reproduce the exact same behavior in other laboratories, even when the training protocol, the experimental hardware, software, and procedures are standardized[1]. In many situations, it is also possible that the behavioral performance is the same, but the strategy used to perform the task is different. Consider, for example, a task in which multiple stimulus properties must be mapped onto appropriate behavioral responses. Such a task can be accomplished by rote learning of this map, but if the task involves structure across stimulus attributes, including irrelevant stimulus features, learning can be

simplified by adopting more "intelligent" strategies that exploit this structure. All these strategies may produce the same level of task performance, so how can we reveal them?

Here, we show that this can be done by examining the geometry of stimulus representations in the state space of recorded neural activities in the dorsolateral prefrontal cortex. The recorded neural responses are typically very diverse and seemingly disorganized[2–5]. However, when the neural activity is analyzed at the population level, it is often possible to identify interesting and informative "structures". The analysis of the geometry of the neural representations has recently revealed that some variables are represented in a special format that enables generalization to novel situations[6]. The representational

[1]Center for Theoretical Neuroscience, Columbia University, New York, NY, USA. [2]Zuckerman Mind Brain Behavior Institute, Columbia University, New York, NY, USA. [3]Center for Neural Science, New York University, New York, NY, USA. [4]SixthFactor Pte. Ltd, Singapore, Singapore. [5]Department of Physiology and Pharmacology, Sapienza University of Rome, Rome, Italy. [6]Department of Neuroscience, Vagelos College of Physicians and Surgeons, Columbia University Irving Medical Center, New York, NY, USA. [7]Kavli Institute for Brain Science, Columbia University, New York, NY, USA. ✉e-mail: vf2266@columbia.edu; aldo.genovesio@uniroma1.it; sf2237@columbia.edu

geometry is defined by the distances between points representing different experimental conditions in the neural activity space. The set of points of all the experiment conditions defines an object with specific computational properties[7], which can be preserved across subjects[8]. For example, if the points define a high-dimensional object (in this article, we always consider the embedding dimensionality[9] when we speak about dimensionality), then a linear decoder can separate the points in a large number of ways, permitting a downstream neuron to perform many different tasks[2,3]. If, instead, the points define a low-dimensional object, the representations allow a simple linear decoder of one variable to generalize across the values of other variables[6]. These representations have been called abstract because of their generalization properties, and they are known as disentangled representations in the machine learning community[10,11]. Abstract representations have been observed in several brain areas[6,12–20], but it is still unclear whether they are linked to behavior. As pointed out by Krakauer et al.[21], a new conceptual framework that meaningfully maps the neural data to behavior is necessary to understand the brain-behavior relationship better, and to accomplish that, the analysis of the behavior should be as fine-grained as the analysis performed on the neural data.

Here, we show that differences between neural representational geometries across subjects reflect significant differences in their behavior, providing evidence that the aspects of the representational geometry we studied here could affect behavior. Our results indicate that the analysis of the representational geometry can be an important tool for reliably interpreting individual differences in behavior.

More specifically, we analyzed the activity of neurons recorded in the dorsolateral prefrontal cortex (PFdl) of two monkeys performing a visually cued rule-based task[22]. The task required choosing between two spatial targets based on the rule cued by a visual stimulus, either staying with the same response as in the previous trial (after a stay cue) or shifting to the alternative response (after a shift cue). Crucially, the average task performance was the same for the two monkeys.

We systematically studied the aspects of the geometry of neural representations that have interesting computational implications[6]. Our analysis revealed that the representational geometry is strikingly different for the two monkeys: the first monkey is more "visual", representing the shape of the visual cues in an abstract format. The second monkey is more "cognitive," representing the rule in an abstract format. This finding brought us to reanalyze the behavior, and we discovered that the reaction time patterns actually reflect the different representational geometries.

We then used Recurrent Neural Network (RNN) models to explain mechanistically these differences. We trained multiple RNNs to perform the same cued rule-based task used in the experiment. Each RNN was randomly initialized and trained on a different sequence of stimuli. The training was interrupted when the RNN reached a certain level of performance. Although all the different RNNs performed equally well, different networks exhibited different representational geometries. Given the same learning stage, the geometries of the networks that reached high performance earlier were similar to those observed in the more "visual" monkey, while the RNNs that learned more slowly exhibited a geometry that resembled the one more "cognitive" monkey. The reaction times of the RNNs and the monkeys exhibited the same patterns.

Our study demonstrates that the analysis of representational geometry enables us to discern individual differences in task-performing strategies. Furthermore, the compelling connection we established between the representational geometry and observed behavior underscores the critical role these geometric aspects might play in the execution of the task.

## Results

We analyzed single-unit recordings in the dorsolateral prefrontal cortex (PFdl) of two male rhesus monkeys. As the main message of this work is that the representational geometry can explain the differences in the behavior of the two monkeys, we will present the neural and behavioral results for each monkey separately. We refer to them as Monkey 1 and Monkey 2.

Both monkeys were trained to perform a visually cued rule-based task (Fig. 1A). The task was to choose one of two spatial targets with a saccadic movement, according to the rule instructed in each trial by one of four possible visual cues (Fig. 1B). Two cues instructed the monkey to "stay" with the target chosen in the previous trial, while the other two cues instructed to "shift" to the alternative target. In each trial, the visual cue was randomly chosen. At the time of the recordings, both monkeys had already been trained, and they performed the task with the same high accuracy.

We found significant differences between the two monkeys when we analyzed the geometry of the neural representations recorded during the task. The representational geometry is defined by the set of distances between the points in the firing rate space that represent different conditions (see ref.[23]). This is a relatively large set of variables, which are not defined in a unique way as there are several reasonable measures of distances in the presence of noise. We focused on three particular aspects of the geometry that have the advantage of being cross-validated and interpretable: the first is the set of linear decoding accuracies for the task-relevant variables and all the other variables that correspond to balanced dichotomies of the conditions (i.e., all the possible ways of dividing the conditions into two equal groups). The task-relevant variables are the previous response, the rule, the current response, and the shape of the visual cue (Fig. 1D). What we defined as shape identifies whether the visual cue is a rectangle or a square, although the cues also differ because the rectangles are gray with different orientations and the squares are colored (yellow and purple). The decoding accuracy is directly related to the distance between two groups of points, and in this respect, it is a geometrical measure. It is better than the average distance because it is cross-validated and it takes into account the structure of the noise, similar to the Mahalanobis distance[24]. Moreover, it is interpretable because it tells us something about the variables that are represented.

The second aspect of the geometry is related to the ability of a linear classifier to generalize across conditions when trained to decode the balanced dichotomies (cross-condition generalization performance or CCGP[6]). For example, consider a representation of a visual stimulus that is characterized by a shape and a color. Shape can be either a triangle or a circle, and color is either red or blue, for a total of 4 different stimuli. In Fig. 2A we show two possible geometries. Each point represents the response of a population of three neurons to one of the four stimuli. The activity of each neuron is represented along a different coordinate axis. The geometry depicted in Fig. 2A-left shows the 4 points arranged on a square (factorized or disentangled representation), where shape and color are encoded along two orthogonal directions. The CCGP for shape is defined as the performance of a linear decoder to report the shape of the stimulus (circle or triangle) when its color is blue (testing set), after it was trained only on red stimuli (training set). For this kind of geometry, the linear decoder trained to classify the red stimuli can readily generalize to blue objects, resulting in high CCGP for variable shape. For this geometry, color also has an elevated CCGP (a decoder trained to report color for triangles would generalize right away to circles). The CCGP of shape depends on the angles between coding directions and in order to generalize to blue objects, it is necessary that the coding direction of shape for red stimuli (i.e., the direction from the points corresponding to neural activities from circle to triangle) is approximately the same for stimuli. If we consider a second kind of geometry where the 4 points are placed at random positions in the activity space (Fig. 2A-right), CCGP is low as a decoder trained to classify the shape only on red objects does not generalize to the blue ones. In general, a variable with high CCGP is encoded in a special format that we define as "abstract"[6]. The variable,

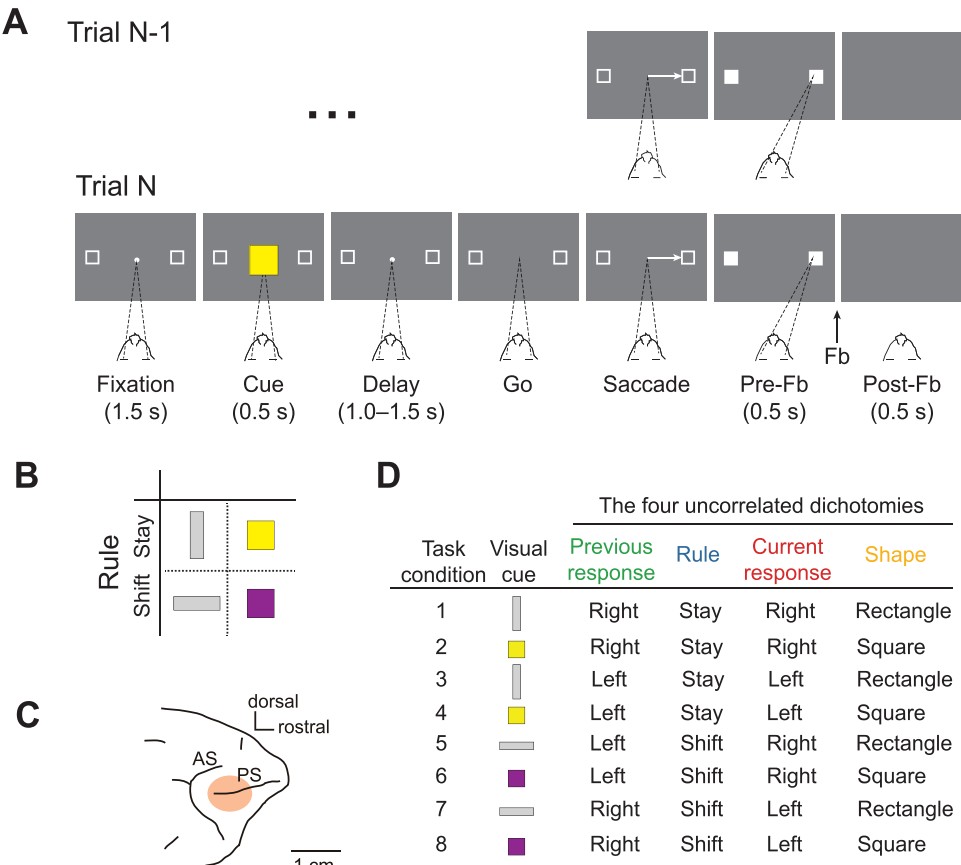

**Fig. 1 | Behavioral task, visual cues, recording site, and task conditions.**
**A** Example of two consecutive trials of the visually cued rule-based task with temporal ordering of task events from left to right. The dark gray rectangle represents the video screen as viewed by the monkey. The target of the monkey's gaze is indicated by dashed lines. In this example, trial N is a stay trial instructed by the yellow square, requiring the monkey to choose the same target (right) chosen in the previous trial N-1. Fb, Feedback. Figure adapted from Fascianelli, V., Tsujimoto, S., Marcos, E. & Genovesio, A. Autocorrelation Structure in the Macaque Dorsolateral, But not Orbital or Polar, Prefrontal Cortex Predicts Response-Coding Strength in a Visually Cued Strategy Task. Cerebral Cortex 29, 230–241 (2017). https://doi.org/10.1093/cercor/bhx321[47]. **B** Visual cues presented to the monkey. Each visual cue instructed the rule to be applied: the vertical gray rectangle and yellow square instructed the stay rule; the horizontal gray rectangle and purple square instructed the shift rule. **C** Recording area in dorsolateral prefrontal cortex. AS, Arcuate Sulcus; PS, Principal Sulcus. Figure adapted from Fascianelli, V., Tsujimoto, S., Marcos, E. & Genovesio, A. Autocorrelation Structure in the Macaque Dorsolateral, But not Orbital or Polar, Prefrontal Cortex Predicts Response-Coding Strength in a Visually Cued Strategy Task. Cerebral Cortex 29, 230–241 (2017). https://doi.org/10.1093/cercor/bhx321[47]. **D** List of the eight task conditions defined as the combination of the four main uncorrelated dichotomies: previous response (green), rule (blue), current response (red), and shape of the visual cue (orange). The color code of the four dichotomies is conserved across all the figures.

i.e. shape, is encoded in an abstract format (or simply, it is abstract) because the coding direction does not depend on the specific instance, i.e., color. CCGP also takes into account the noise structure, and it is cross-validated.

The third aspect of the geometry is the dimensionality of the representation, which we assessed by using the shattering dimensionality. The shattering dimensionality is defined as the average linear decoding performance for all possible balanced dichotomies. A high shattering dimensionality means that the linear decoder can separate (shatter) the points in any possible way, enabling a linear readout to perform a large number of input-output functions, like the Exclusive OR (XOR) configuration, that wouldn't be possible in case of low shattering dimensionality (see Fig. 2B)[2,6].

**Differences between the neural representational geometries of the two monkeys**
We recorded 289 neurons from Monkey 1 and 262 neurons from Monkey 2 in PFdl (Fig. 1C). To investigate which task variables can be decoded, we built pseudo-simultaneous trials (pseudo trials) for each monkey separately (see Methods). We defined the pseudo trial as the

combination of spike counts randomly sampled from different trials of the same task condition[2]. For each neuron, the spike count was estimated in a $200 ms$ time bin. We considered only neurons recorded for at least 5 complete and correct trials (not correction trials) in each task condition for a total of 205/289 (71%) neurons for Monkey 1 and 188/262 (72%) neurons for Monkey 2.

We found that in Monkey 1, almost all the dichotomies can be linearly decoded during the cue presentation (Fig. 3A). This is not surprising, as it has already been observed in cognitive areas of monkeys[2,6]. It is typically the expression of the high dimensionality of the neural representations (for the maximal dimensionality, all dichotomies are linearly decodable) and a consequence of the diverse mixed selectivity responses of the neurons. However, this does not necessarily mean that the representations are completely unstructured. Indeed, in ref.[6], high decodability coexisted with elevated CCGP for a few variables, which instead indicates low dimensionality. This is the case also in our dataset, in which we observed a much smaller number of dichotomies to be in an abstract format, i.e., with a high CCGP (Fig. 3B). In Monkey 1, shape is the variable with the highest CCGP, in the time interval right after the presentation of the

stimulus. Later, during the trial, the current response is the most strongly encoded variable with the largest CCGP. The previous response and the rule can be decoded but their CCGP is at chance, and hence these variables are not in an abstract format. It is worth noting that the rule is represented in an abstract format in a later period, after the cue offset (Fig. 3B). In Monkey 2, almost all the dichotomies can be linearly decoded during the cue presentation, except for the shape and the previous response (Fig. 3C). The CCGP analysis reveals that, in Monkey 2, the rule is in an abstract format with the highest CCGP during the cue presentation, differently from Monkey 1 (Fig. 3D). In both monkeys, instead, during the cue presentation, the current response is in an abstract format, while the previous response is not.

We performed an additional analysis to control whether the main differences in decoding accuracies between monkeys could depend on differences in the recording sites. Supplementary Fig. 1A shows the location of the penetrations of the recordings in PFdl for both monkeys. It reveals an overlap between the recording sites in the two monkeys, particularly in the dorsal region to the principal sulcus, where all neurons in Monkey 2 were recorded. Thus, we examined the dorsal (106 neurons) and ventral (99 neurons) recordings separately for Monkey 1. When comparing the decoding accuracy of the task variables between neurons in the dorsal sites only (which match exactly with the sites in Monkey 2) with the combined dorsal and ventral recordings (see Fig. 3A and Supplementary Fig. 1B), we found comparable results. For this reason, we combined the recording sites in Monkey 1 in all the following analyzes. The main difference between the dorsal and ventral recordings regards the representation of the previous response, which is encoded mainly in neurons in the ventral sites. This matches the previous response signal in Monkey 2, which is weakly decoded during the fixation period. It might be possible that even Monkey 2 could have a stronger previous response representation ventrally to the principal sulcus, but we lack the ventral recordings.

To better highlight the differences in the representational geometry, we focused on the 300 ms time window in which the differences are large (from 200 ms after the cue onset until the cue offset, gray vertical shade in Fig. 3). The beeswarm plots in Fig. 4A show the decoding accuracy and CCGP for all the possible dichotomies in the 300 ms time window for Monkey 1 and Monkey 2. It is evident that Monkey 1 represents the shape of the visual cue in an abstract format (highest CCGP), while the rule is not abstract (CCGP at chance), even though it can be decoded (Fig. 4A). The rule becomes abstract only later in the delay period after the cue offset (Fig. 3B). Instead, for Monkey 2, the rule is the variable with the highest CCGP, while the shape of the visual cue is not abstract (CCGP at chance) (Fig. 4A). Moreover, both monkeys represent the current response in an abstract format but not the previous response. Interestingly, in both monkeys, the current response is not abstract from the time when it can be decoded, but only slightly later (see Fig. 3). These results suggest that Monkey 1 is grouping together the cues with the same shape, and hence, it might be using a strategy based on the identity of individual visual stimuli. Instead, Monkey 2 might be using a more "cognitive" strategy because the rule is the variable with the highest decoding accuracy and CCGP, and hence Monkey 2 is grouping together the visual cues that correspond to the same rule, even though they are visually very different.

We also assessed the dimensionality of the representations by measuring their shattering dimensionality (SD). We observed a higher SD in Monkey 1 than in Monkey 2 during the 300 ms time window in which the differences in representational geometries are large (Fig. 4A). A higher SD in Monkey 1 might be compatible with implementing a more "visual" strategy based on a lookup table, for which each visual cue is uniquely associated with a mapping from the previous response to the current response.

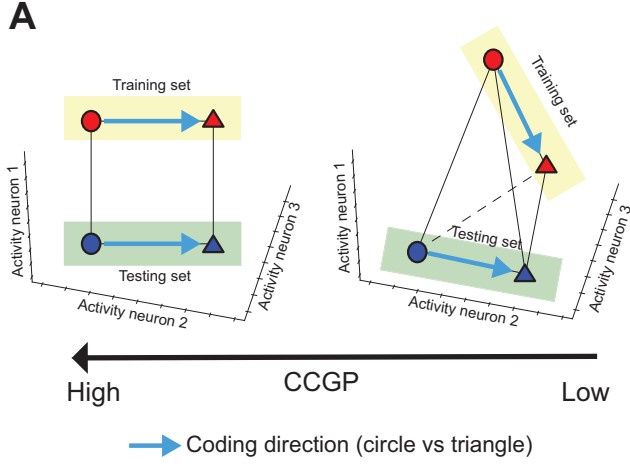

**A**

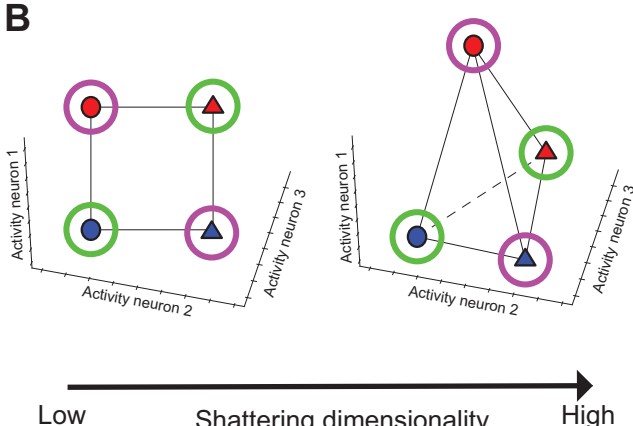

**B**

**Fig. 2 | Schematic of different representational geometries for 4 conditions in the neural activity space and their properties. A** Left: factorized or disentangled representations where the 4 points are arranged on a square. The shape (circle vs triangle) and color (red vs blue) are encoded along two orthogonal directions. This geometry supports the representation of shape (and color) in abstract format, i.e., high CCGP. Right: Random representation where the 4 points are placed at random locations in the activity space. This geometry does not support the representation of the shape in abstract format, i.e., low CCGP. **B** Left: Low shattering dimensionality, where the 4 points are placed at the vertices of a square. The shattering dimensionality is low because not all the dichotomies can be decoded by a linear decoder due to the XOR configuration (purple and green circles). Right: High shattering dimensionality supports the decoding of a higher number of dichotomies, including the one not linearly decodable, i.e., XOR.

Since the shape cannot be decoded in Monkey 2 using a linear classifier, we were wondering whether it is not encoded at all or if it could be decoded using other decoders. We decided to consider pairs of conditions separately, which is equivalent to considering non-linear decoders for all the points. Indeed, if two conditions are sufficiently separated, i.e., the distance between the corresponding points is large enough compared to the noise, then a linear decoder should work. This is true even when the dichotomy is not linearly separable. For example, in the case of XOR for four points that define a low-dimensional object like a square, a linear decoder would not be able to separate the two points on the diagonal from the other two, but it would separate all pairs of points if taken one pair at the time. In addition to considering pairs of points, we denoised the data by projecting the neural activity of a single pseudo trial into a lower-dimensional space (3D) using the Multi-Dimensional Scaling (MDS) technique described in the Methods. Using this procedure, we found

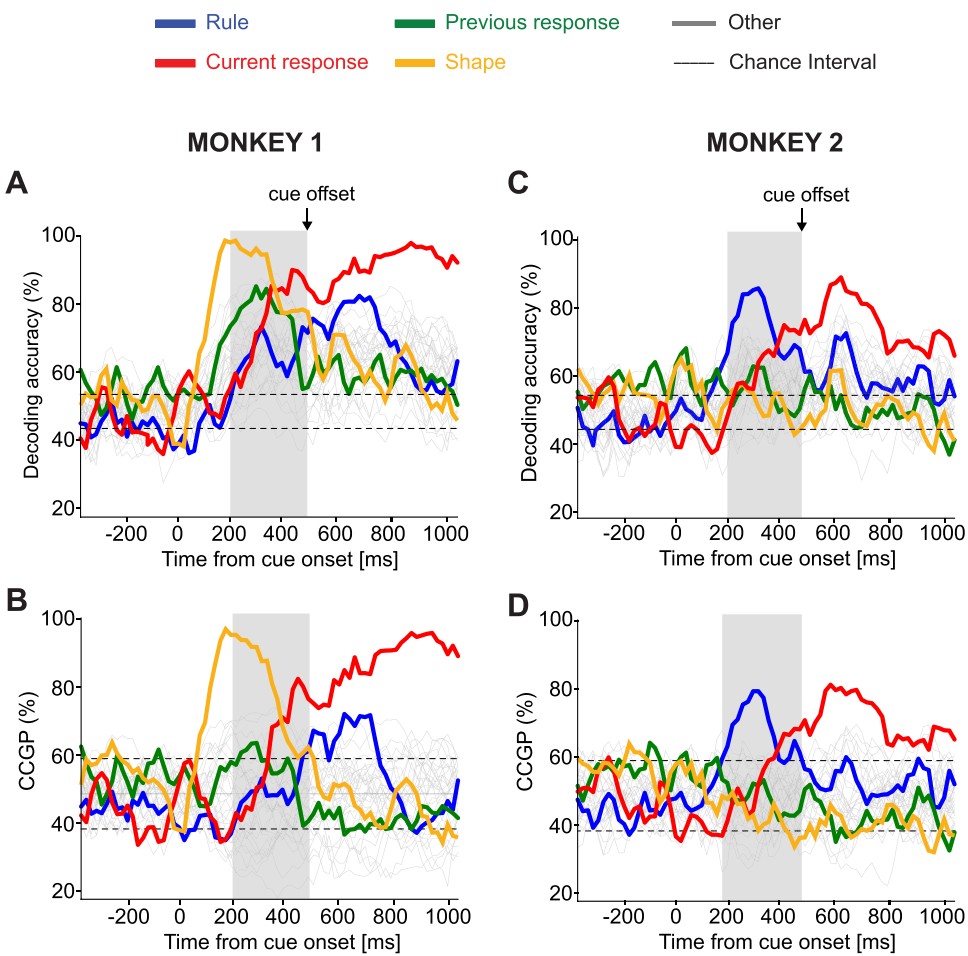

**Fig. 3 | Decoding accuracy and CCGP as a function of time.** Time is aligned to the cue onset lasting for 500 ms (until the time of the cue offset indicated by the vertical black arrow). The horizontal dashed lines are ± 2 standard deviations of 100 cross-validations distribution obtained from null models. The gray vertical shade indicates the time bin starting at 200 ms after cue onset until cue offset, in which we found the maximal difference between the neural representations of the two monkeys. **A** Decoding accuracy of all the possible 35 dichotomies (i.e., all variables that correspond to grouping the conditions into two equal-size groups) in Monkey 1. During the cue presentation, most of the dichotomies can be decoded, in particular all the main task variables indicated with different colors. The shape of the visual cue (orange) is decoded with the highest accuracy, followed in time by the previous response (green), the current response (red), and the rule (blue). **B** CCGP of the 35 dichotomies in Monkey 1. During the cue presentation, the shape (orange) is in an abstract format with the highest CCGP, followed in time by the current response (red). The rule (blue) is not abstract during the cue presentation, but it becomes abstract after the cue offset. The previous response (green) is not in an abstract format. **C** Decoding accuracy of all dichotomies in Monkey 2. During the cue presentation, the rule (blue) and the current response (red) can be significantly decoded. **D** CCGP of all dichotomies in Monkey 2. Differently from Monkey 1, the rule (blue) is in an abstract format with the highest CCGP during cue presentation, followed in time by the current response (red). The shape (orange) and previous response (green) are not in an abstract format. Source data are provided as a Source Data file.

that the shape can be decoded in both monkeys. In particular, in Monkey 1, the shape can be decoded for both rule conditions with high accuracy (Fig. 4B). This was expected as the shape was already linearly decodable for all the points without denoising (decoding accuracy in Fig. 4A, left). Shape could also be decoded in Monkey 2, in both rule conditions (Fig. 4B). These results show that both monkeys' PFdl neurons encode the shape of the visual stimulus but with different geometries, making the shape linearly separable and in an abstract format only in one of the monkeys.

To visualize the different geometries of the two monkeys, we used the MDS transformation to reduce the dimensionality of the original representations. The MDS preserves the pairwise distances between data points. This is crucial to study the representational geometry in terms of the relative distances between points (task conditions) in the firing rate space. Indeed, MDS ensures that similar neural activity patterns in high-dimensional space remain close to each other in the lower-dimensional representation[6]. More specifically, we used MDS on the dissimilarity matrix containing the Euclidean distances between

the average activity of two task conditions normalized by the variance along the direction that goes from one condition to the other (see Methods). Each point in the MDS plots is the average firing rate of each task condition in a 300 ms time window during the cue presentation (Fig. 5). For each monkey, we highlighted the different dichotomies (groups of conditions) by drawing lines between the conditions that are in the same group. In particular, the shape of the visual cue and current response are in an abstract format in Monkey 1, while the rule and current response are abstract in Monkey 2. For both monkeys, the current response is in an abstract format, while none have the previous response in an abstract format.

**Behavioral differences between monkeys reflect differences in the neural representational geometry**

The differences in the representational geometry are so striking that they induced us to reanalyze the behavior to look for more subtle individual differences. We analyzed 65 and 77 sessions for Monkey 1 and Monkey 2, respectively. As mentioned, we did not find any

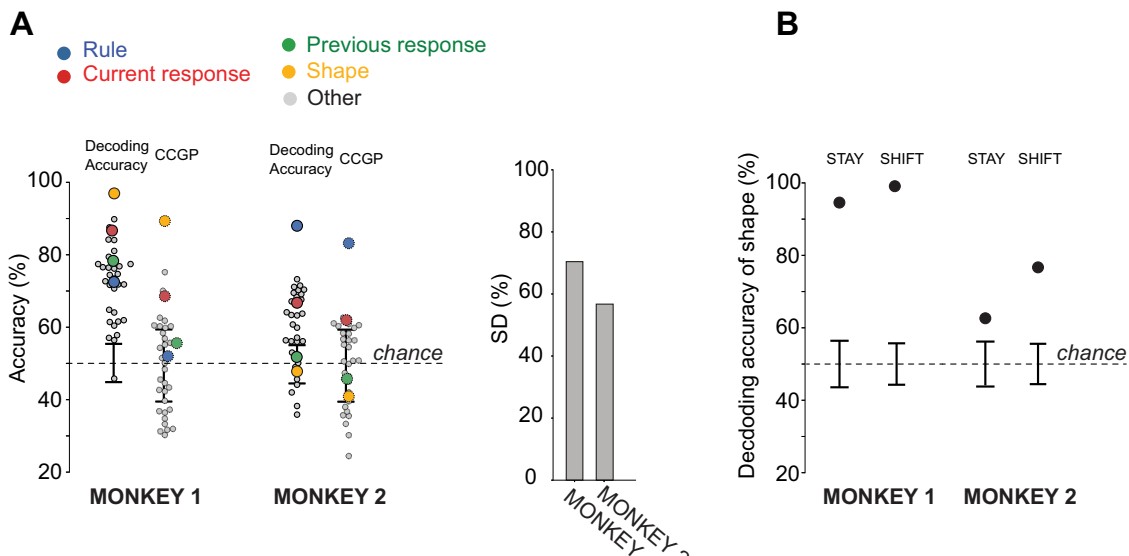

**Fig. 4 | Summarizing the features of the representational geometry: decoding accuracy, CCGP, and shattering dimensionality (SD) during the last 300ms of cue presentation. A** Left: decoding accuracy (continuous-edge circles) and CCGP (dashed-edge circles) for each of the 35 dichotomies in the 300 ms time bin during cue presentation in Monkey 1 and Monkey 2. Each circle is a different dichotomy. The four main dichotomies corresponding to task variables are highlighted with different colors. All the other dichotomies are in gray. Black error bars are the ± 2 standard deviations around the chance level (50%) obtained from 100 null models. Right: SD for Monkey 1 and Monkey 2 in the 300 ms time bin during the cue

presentation. The dimensionality of neural representation is higher in Monkey 1 than in Monkey 2. **B** Decoding accuracy of shape in Monkey 1 and Monkey 2 in the stay and shift rule. The linear decoder was trained after projecting the neural activity of each pseudo trial in a lower-dimensional space using a Multi-Dimensional Scaling (MDS) transformation. The shape is decodable in both rule conditions in both monkeys, though the accuracy is lower in Monkey 2. Black error bars are the ± 2 standard deviations around the chance level obtained from 100 null models. Source data are provided as a Source Data file.

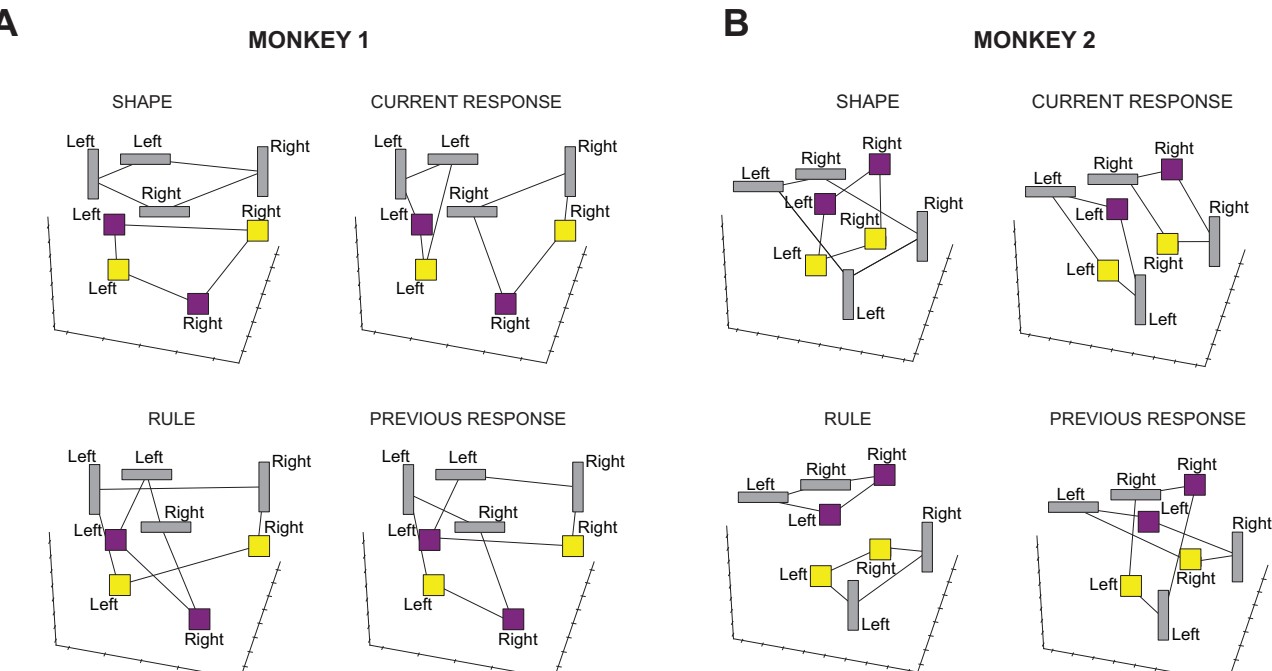

**Fig. 5 | 3D Multi-Dimensional Scaling (MDS) plots.** Each point represents the average firing rate in the 300 ms time window during the cue presentation for one of the eight task conditions. The visual cue is shown along with the current response. The connecting black lines highlight the dichotomy indicated in the label above each plot. Shape and rule are the two dichotomies that mostly characterize the difference in representational geometry between the two monkeys, while the

current response is in an abstract format for both monkeys. The previous response is not abstract in either of the monkeys. **A** 3D MDS plots in Monkey 1 for shape (top-left), current response (top-right), rule (bottom-left), and previous response (bottom-right). **B** The same as in **A** for Monkey 2. Source data are provided as a Source Data file.

significant difference in the overall behavioral performance between the two monkeys (chi-square test, $p$-value = 0.93; Fig. 6A left). However, a significant difference emerged in the average reaction times (Mann–Whitney U test, $p$-value = $10^{-15}$; Fig. 6A right). We then decided to analyze the behavior with finer-grained analyzes. Indeed, since the neural analysis revealed a difference in the representational geometry of the shape and rule between the monkeys, we computed the average behavioral performance for each condition separately by grouping the correct trials according to shape (rectangle and square) and rule (stay and shift). There is not a significant difference in the behavioral performance between different shapes (chi-square test: $p$-value = 0.78 in Monkey 1, Fig. 6B left; $p$-value=0.06 in Monkey 2, Fig. 6D left), and between different rules (chi-square test: $p$-value = 0.11 in Monkey 1, Fig. 6B right; $p$-value = 0.22 in Monkey 2, Fig. 6D right) in both monkeys. Nevertheless, a significant difference in reaction times emerged across conditions in each monkey. In particular, Monkey 1, with high decoding accuracy and CCGP for the shape, has an average reaction time that significantly changes with the shape of the visual cue (Mann–Whitney U test: $p$-value = 0.002; Fig. 6C, left) regardless of the rule (Mann–Whitney U test: $p$-value = 0.05; Fig. 6C, right). On the opposite, Monkey 2, with high decoding accuracy and CCGP for the rule, shows an average reaction time that significantly changes with the rule (Mann–Whitney U test: $p$-value = $10^{-10}$; Fig. 6E right) regardless of the shape (Mann–Whitney U test: $p$-value = 0.28; Fig. 6E left).

The differences in reaction times are significant, and they nicely reflect the representational geometry, but they are relatively small. So we decided to investigate further the behavior to see whether these differences could be predicted by looking at the recent series of events and monkey responses. In particular, we fitted a multi-linear regression model to predict the reaction time on a trial-by-trial basis using three factors: the previous response, the shape of the visual cue, and the rule. We also considered all the interaction terms (see Supplementary Fig. 2). We found that the rule factor has a stronger weight in predicting reaction times in Monkey 2 than in Monkey 1 (Mann–Whitney U test: $p$-value = $10^{-34}$; Fig. 6F). Vice versa, the shape is a stronger factor in predicting the reaction time of Monkey 1 (Mann–Whitney U test: $p$-value = $10^{-34}$; Fig. 6F). Supplementary Fig. 2 shows that the strongest factor in predicting the reaction time is the interaction of the previous response and the rule in both monkeys because the combination of these two factors is essential for choosing the correct response. Indeed, both monkeys have been trained to combine the previous response and rule to provide the current correct response.

## Correlation between representational geometries and reaction times in recurrent neural networks

To better investigate the differences in representational geometries between the two monkeys and the relation with reaction times, we trained Recurrent Neural Networks (RNNs) to perform the visually cued rule-based task through deep reinforcement learning algorithms[25]. Artificial neural networks have been shown to be a powerful tool for understanding the normative aspects of neural representations[26], especially when trained with reinforcement learning, which resembles the protocols used to train animals[27,28]. As these networks are often over-parametrized, the same task performance is often obtained with numerous different choices of network parameters. When these networks are trained multiple times, the solutions found by the same training algorithm can substantially differ, leading to networks that might implement qualitatively different strategies to solve the same problem. This is usually considered an inconvenience, as much as the individual differences observed across animals. In our case, we took advantage of this variability across simulations to understand the relation between different geometries and the reaction times that were observed in the experiments.

The inputs we provided to the network were: (1) the visual cue, encoded by two one-hot vectors of three units each, with the

first vector encoding the shape and the second vector the color; (2) the previous response, encoded by a one-hot vector of two units; (3) the fixation input, a scalar that instructs the network either to maintain fixation or not (Fig. 7A). The inputs were passed through fixed, non-trained, random weights to an expansion layer of 100 Rectified Linear units. The output of the expansion layer was then processed by 100 recurrent units in a time-discretized vanilla RNN architecture, where the network provided outputs at each time step. The readouts included a scalar representing the temporal discounted expected return, defined as the output of the value function (critic), and a real vector with a length equal to the possible actions (actor policy). The action at each time step was determined by sampling from the categorical distribution obtained from the softmax of this vector (for examples of two trials, see Supplementary Fig. 3A, B).

We trained 80 RNNs using the proximal-policy-optimization (PPO) as a deep reinforcement learning algorithm[25], and each network was trained on a different random sequence of trials. The architecture of the network, the learning algorithm, and the input statistics were the same for all the RNNs. However, the networks' weights and biases were initialized randomly, using a different realization for every network. The temporal structure of the task is the same as shown in Fig. 1A, except for the pre- and post-feedback periods, which were not implemented in the model since they are not of interest in the current study; the trial type was drawn randomly at the beginning of each trial during training. Each RNN was exposed to a different random sequence of trials. We stopped the training when two conditions were satisfied: (1) the network achieved at least 99% of complete trials, i.e., trials in which the decision was made within an interval of 1500 ms from the presentation of the visual stimulus (see Materials and Methods); (2) at least 90% of correct trials out of the complete ones on a validation batch of 10000 trials (Supplementary Fig. 4A, B). So, all the RNNs that we studied performed the same. However, the number of learning epochs to reach that performance was different for different networks.

After terminating the training, we analyzed the activations/firing rates of the recurrent units on a separate testing set comprised of 10000 trials for each network. Our model aimed to explore the potential relation between the representational geometries of shape and rule and the corresponding reaction times. To quantify this relation, we first had to characterize the differences in the geometry across different RNNs. We defined a variable, Δ-decoding, which is the difference in the decoding accuracy of rule and shape during the cue presentation period. We also defined the variable Δ-CCGP, which is defined in the same way as Δ-decoding but for CCGP. These two variables provide a simple description of relevant differences in the representational geometry. Indeed, positive values of Δ-decoding indicate that shape is more strongly encoded than rule, resembling the representation observed in Monkey 1 (Fig. 3A, B). Conversely, negative values indicated a stronger signal of rule compared to shape, resembling the representation observed in Monkey 2 (Fig. 3C, D).

We evaluated the Δ-decoding (and CCGP) for each trained RNN and examined its correlation with the number of training trials needed by each network to reach the stop-learning criterion. We studied the RNNs along a continuum spectrum of training trials, ranging from tens of thousands to hundreds of thousands. Importantly, all these networks had the same performance. Our analysis revealed a significant negative correlation between Δ-decoding and the training duration, as depicted in Fig. 7B (Pearson coefficient, $\rho = -0.52$, $p$-value = $10^{-6}$). Similarly, the Δ-CCGP exhibited the same trend, as shown in Fig. 7C (Pearson coefficient, $\rho = -0.48$, $p$-value = $10^{-5}$).

These findings suggest that the networks that reached the performance threshold with fewer training trials developed a stronger shape representation, as indicated by higher decoding accuracy (see Supplementary Fig. 5). Interestingly, these networks also exhibited a

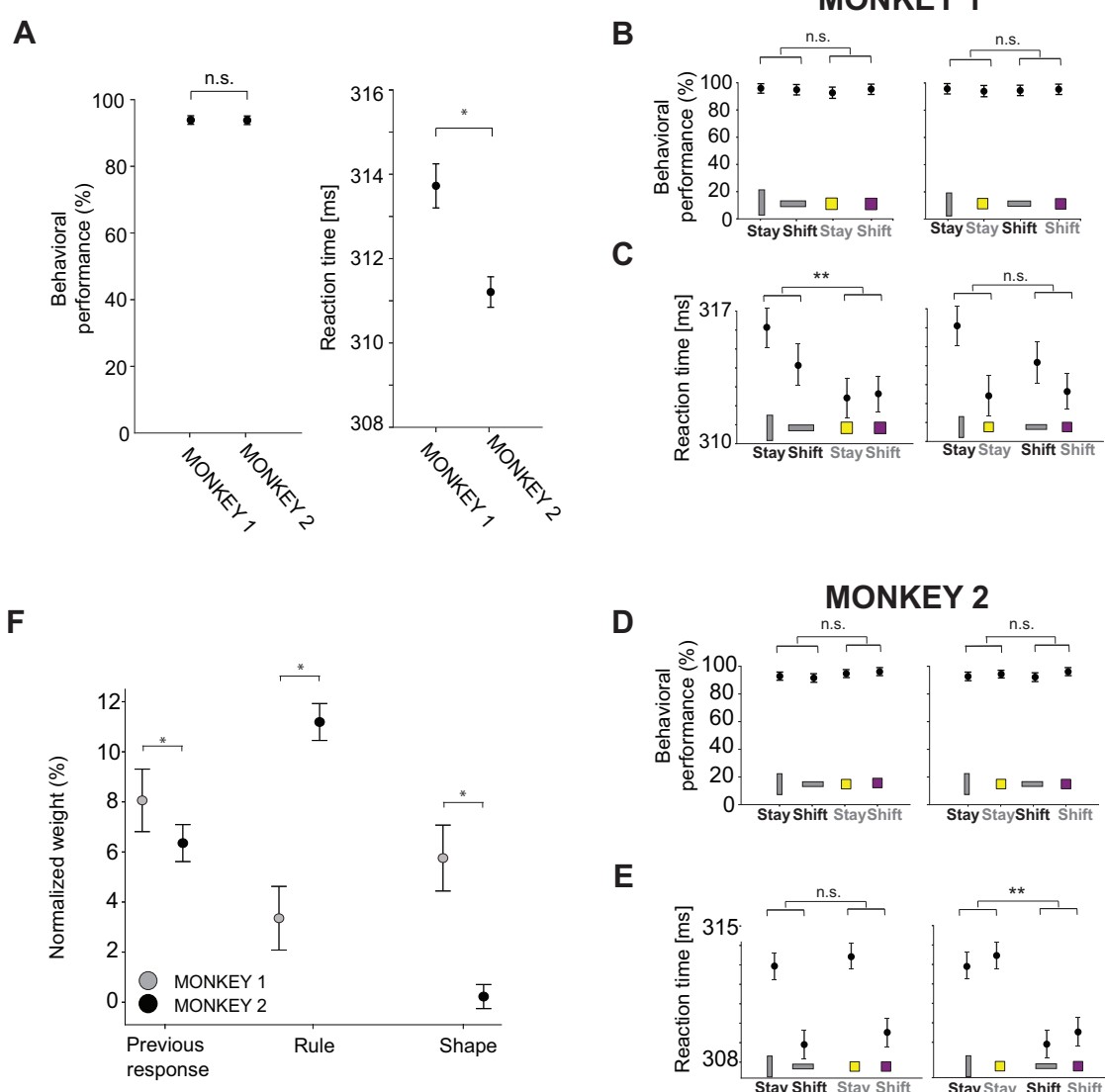

**Fig. 6 | Behavioral performance, reaction times of the monkeys, and multilinear regression behavioral model reflecting differences in the neural representational geometries. A** Left: average behavioral performance across sessions for Monkey 1 ($n = 65$ sessions) and Monkey 2 ($n = 77$ sessions). Both monkeys performed the task with high accuracy. The error bars indicate the confidence interval at 95% of confidence level. Right: average reaction time across sessions for Monkey 1 ($n = 65$ sessions) and Monkey 2 ($n = 77$ sessions). A significant difference emerged in the average reaction time between the two monkeys. The error bars are the standard error of the mean. n.s. not significant: chi-square test, $p$-value = 0.93. *: Mann−Whitney U test, two-sided, $p$-value = $10^{-15}$. **B** Mean behavioral performance across sessions for Monkey 1 ($n = 65$ sessions) computed separately for each rule and shape. The x-axis indicates the rule, and the y-axis is the mean performance averaged across sessions. The visual cue of each condition is indicated at the bottom of the plot. On the left(right), the visual cue order reflects shape(rule). n.s. not significant (left, chi-square test, $p$-value = 0.78; right, chi-square test, $p$-value = 0.11). **C** Mean reaction time across sessions for Monkey 1 ($n = 65$ sessions). As in

**B**, the x-axis indicates the rule, and the y-axis is the reaction time averaged across sessions. The error bar is the standard error of the mean. n.s. not significant: Mann−Whitney U test, two-sided, $p$-value = 0.05; **: Mann−Whitney U test, two-sided, $p$-value = 0.02. **D** The same as in **B** but for Monkey 2 ($n = 77$ sessions): n.s. not significant (left, chi-square test, $p$-value = 0.06; right, chi-square test, $p$-value = 0.22). **E** The same as in **C** but for Monkey 2 ($n = 77$ sessions): n.s. not significant: Mann−Whitney U test, two-sided, $p$-value = 0.28; **: Mann−Whitney U test, two-sided, $p$-value = $10^{-10}$. **F** Weights, averaged across 100 models, of the three independent factors predicting the reaction time of single trial in a multi-linear regression model. Each weight is normalized to the total sum of the weights (see Supplementary Fig. 2).The error bars are the 2 standard deviations of weights across 100 models. The variance explained ($r$-squared) by the models is 12% and 18% for Monkey 1 and Monkey 2, respectively. *: Mann−Whitney U test, two-sided, previous response ($p = 10^{-32}$); rule ($p = 10^{-34}$); shape ($p = 10^{-34}$). Source data are provided as a Source Data file.

larger CCGP for shape. The CCGP was generally high, indicating that these representations were abstract, as in the experimental observations. Previous studies demonstrated that artificial simple feedforward neural networks can easily generate this type of abstract representations using backpropagation and reinforcement learning algorithms[6,29].

The networks with higher CCGP for shape quickly found a policy to solve the task up to criterion while maintaining residual shape

coding that comes from network initialization (the expanded inputs retain the low dimensional structure of the disentangled representations of shape and color). Conversely, the RNNs requiring more extensive training exhibited higher decoding accuracy and CCGP for the rule compared to the shape (Supplementary Fig. 5). This result is not surprising because if the networks do not converge right away, the training is more extensive, and the representations tend to inherit the low dimensional structure of the output (see[29] for feed-forward

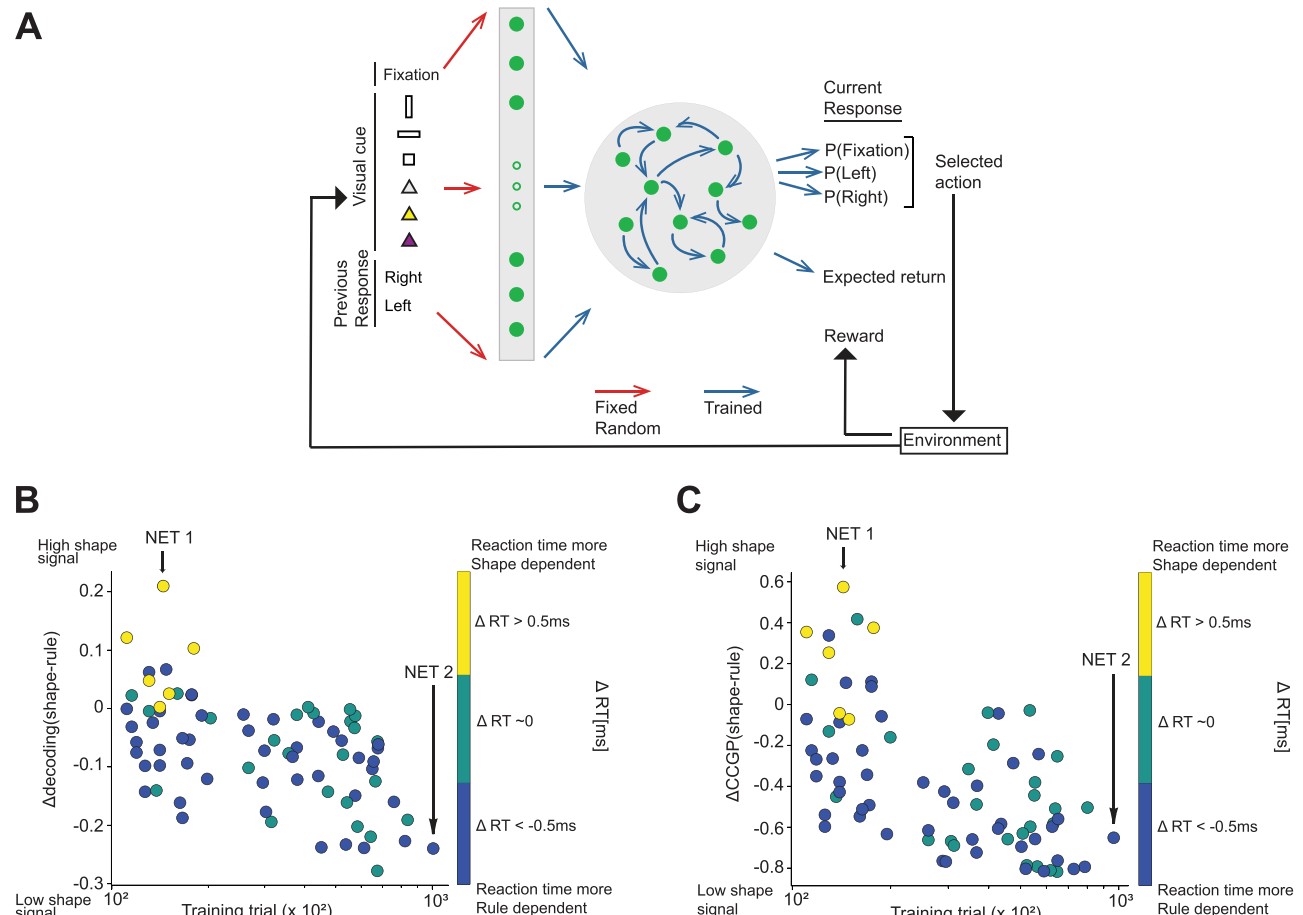

**Fig. 7 | Individual differences across 80 RNNs trained to perform the visually cued rule-based task.** **A** Architecture of the RNN with a schematic of the reinforcement learning setting (agent-environment interaction loop[66]). The inputs consist of one unit for fixation, six units encoding visual cues (vertical rectangle, horizontal rectangle, or square combined with color, gray, yellow, or purple, denoted by triangles), and two units encoding the previous response (right or left). The input is passed through 100 Rectified Linear units. The output of the expansion layer is processed by 100 recurrent units. At each time step, the network gives an output corresponding to fixate or select either right or left, represented by three units (network policy/actor), and the expected discounted return at that time step up to the end of the trial (value function/critic). **B** Decoding results for each RNN illustrating the difference in decoding accuracy of the shape and rule as a function of the number of training trials to reach the performance criterion. The color bar indicates the difference in reaction time between shape and rule (ΔRT): positive values indicate a reaction time influenced by shape irrespective of the rule; negative

values suggest a dependence on the rule regardless of the shape. Significant negative correlation is observed between the difference in decoding the two task variables and the amount of training (Pearson coefficient, $\rho = -0.52$, p-value = $10^{-6}$). Conversely, networks with a lower training requirement exhibit a stronger shape signal. The difference in decoding also shows a significant positive correlation with the difference in reaction time (Pearson coefficient, $\rho = 0.35$, p-value = 0.001). The vertical black arrows indicate two example RNNs with high decoding accuracy for the shape (NET 1) and for the rule (NET 2). **C** This is similar to panel **B**, where the y-axis shows the difference between the CCGP for shape and rule. Significant negative correlation was observed between the difference in CCGP of the two task variables and the amount of training (Pearson coefficient, $\rho = -0.48$, p-value = $10^{-5}$), while a significant positive correlation was observed with the difference in reaction time (Pearson coefficient, $\rho = 0.39$, p-value = $10^{-4}$). Source data are provided as a Source Data file.

networks), combining it with the structure in the inputs. These representations encode less strongly the shape, which is irrelevant for performing the task, and more strongly the rule, which is a combination of input features and their semantics. Thus, our results show that different RNNs, trained to solve the same task with high accuracy above 90%, exhibit distinct representational geometries for specific task variables, influenced by the number of training trials required to attain the same performance threshold.

We then investigated the relation between the representational geometries of shape and rule with the reaction time: would the RNNs exhibiting higher decoding accuracy and CCGP for shape display a reaction time dependent on shape only? Analogously: would RNNs, with higher decoding accuracy and CCGP for the rule, exhibit a reaction time most strongly dependent on the rule? We compared the average difference in reaction time (RT) between trials with different shapes of visual cues to the average difference in reaction time

between trials with different rules. Briefly, this comparison yielded a variable termed ΔRT, which was assigned to each RNN (see Methods). Positive values of ΔRT indicated that, on average, reaction times were influenced by the identity of the shape, regardless of the rule. Conversely, negative values suggested that reaction times were influenced by the rule, regardless of the shape. It is worth noticing that we are interested in studying the patterns of reaction times, regardless of their value in each condition. What we observed to be preserved in the simulations is the grouping of the conditions. So, for example, for those networks with positive values of ΔRT, the two conditions with rectangles have approximately the same average RT, and the two conditions with squares exhibit a different average RT. The RTs for the rectangles are larger than the RTs for the squares in some RNNs. In others, it is the opposite: the RTs for the rectangles are smaller than the RTs for the squares. Our analysis focuses on the patterns of RTs, and ignores which of the two groups of similar RTs is larger than the other.

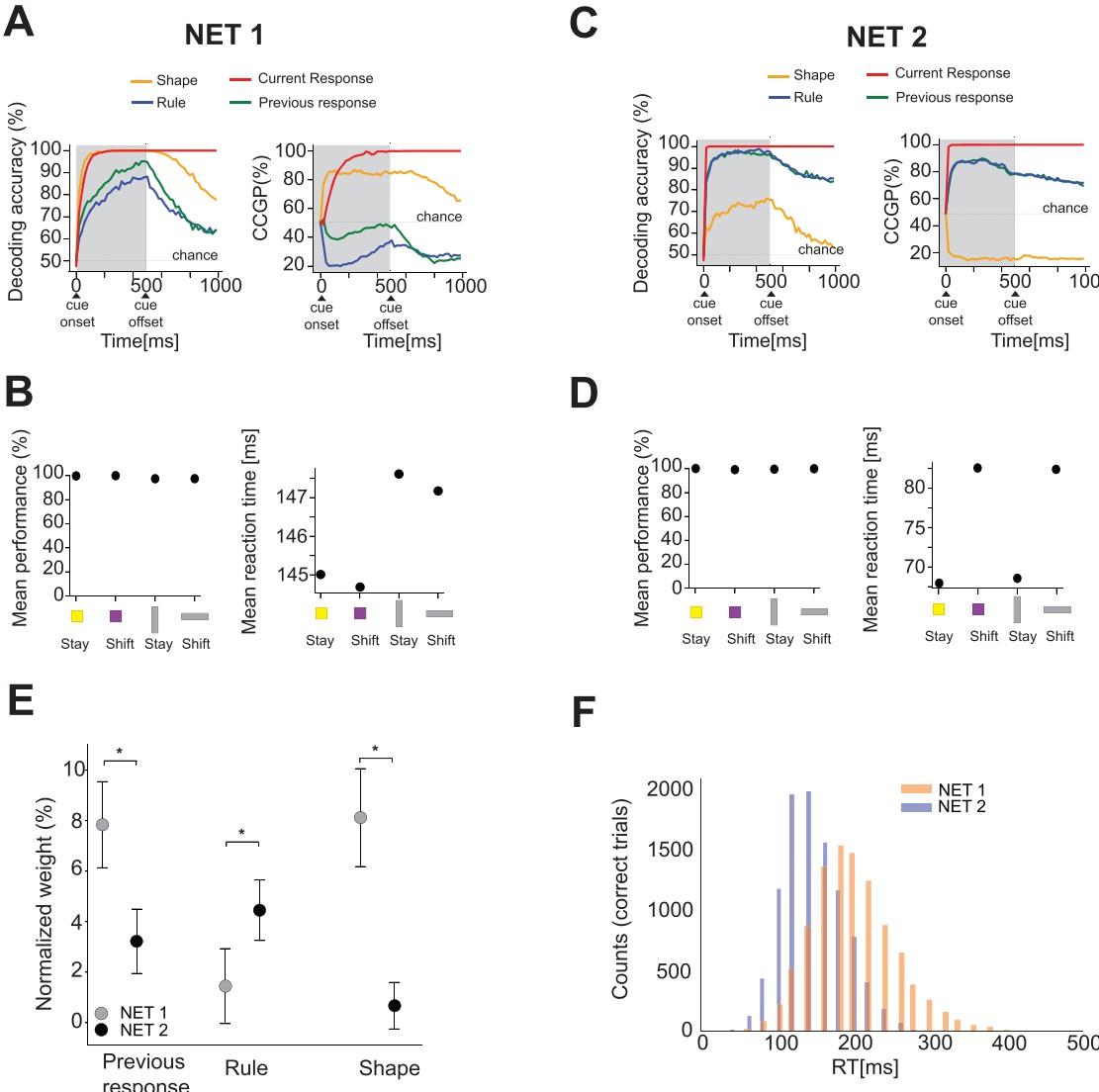

**Fig. 8 | Representational geometries of task variables, behavioral performance, and reaction times of two example recurrent neural networks (RNNs). A** Left: decoding accuracy of the four main uncorrelated variables for the first example RNN (NET 1 in Fig. 7B). The activity is readout from the recurrent units. The shaded area indicates the period where the stimulus is on.. After cue onset, the shape (orange line) is highly decoded, followed by the current response, previous response, and rule. Right: CCGP over time for the four variables. Shape and current response are represented in an abstract format after the cue presentation. **B** Left: average behavioral performance for each condition displayed on the x-axis. Performance exceeds 90% for all conditions. Right: average reaction time for each condition. The representational geometry and reaction time patterns are similar to those of Monkey 1. **C** Left: decoding accuracy of the four main uncorrelated task variables for the second example RNN (NET 2 in Fig. 7B). After cue onset, rule, current, and previous responses are highly decoded. Shape is decoded with lower

accuracy compared to the other variables. Right: CCGP along time for the four variables. The rule, current, and previous responses are represented in an abstract format after cue presentation, while the shape is not. **D** Left: average behavioral performance for each condition displayed on the x-axis. Performance exceeds 90% for all conditions. Right: average reaction time for each condition. This RNN exhibits representational geometry and reaction time patterns similar to those of Monkey 2. **E** Mean of the distribution of the weights of 100 multi-linear regression models. Each weight is normalized to the total sum of the weights (Supplementary Fig. 7A). The error bars are the 2 standard deviations of weights across 100 models. The variance explained (r-squared) by the models is 14% and 30% for NET 1 and NET 2, respectively. *: Mann–Whitney U test, two-sided, previous response ($p = 10^{-30}$); rule ($p = 10^{-34}$); shape ($p = 10^{-34}$). **F** Distribution of RTs for NET 1 (orange) and NET 2 (blue). The mean of the distribution of RTs for NET 2 is smaller than the mean RTs in NET 1. Source data are provided as a Source Data file.

We discovered a significant positive correlation between Δ-decoding and ΔRT across all the RNNs (Fig. 7B, Pearson coefficient, $\rho = 0.35$, p-value = 0.001). Similarly, a significant positive correlation was observed between Δ-CCGP and ΔRT (Fig. 7C, Pearson coefficient, $\rho = 0.39$, p-value = $10^{-4}$). These results indicate that RNNs with a stronger shape signal during cue presentation typically require a shorter training period and exhibit average reaction times that change based on the identity of the shape, regardless of the rule. To gain further insights, we selected, among all the networks, the two ones with the highest shape and rule signals, respectively, and we analyzed

them in detail. Specifically, we computed the decoding accuracy and CCGP over time from stimulus onset for the four uncorrelated task variables, as well as the behavioral performance and average reaction time.

Figure 8 A illustrates the results for one of the examples RNNs, referred to as NET 1, which shows the highest shape decoding accuracy and CCGP, and it reaches the performance criterion with a small number of training trials among all the RNNs depicted in Fig. 7B, C (see Supplementary Fig. 6 for the decoding accuracy and CCGP along time for all the RNNs). During the presentation of the stimulus, the shape

variable exhibits a high decoding accuracy, followed in time by the current response, previous response, and rule. The shape and current response are also represented in an abstract format, with high CCGP, while the rule and previous response are not. Notably, although the rule and previous response can be decoded, they are not represented in an abstract format. Moreover, this network displays high behavioral performance for each task condition (>90%, Fig. 8B, left). Interestingly, the average reaction time varies depending on the shape of the cue, regardless of the rule. Specifically, the network responds faster to square cues compared to rectangle cues, irrespective of the rule (Fig. 8B, right). Other networks that operate in the same regime also exhibit reaction times that depend only on the shape of the stimulus (not shown), but the RT for the squares is longer than for the rectangles. This is not surprising, given that the different visual input features (shape and color) are represented in the same way. It is possible that in the monkey brain, the colored stimuli elicit a more prominent response, and hence, the reaction time would always be shorter for squares than for rectangles. This is something we can easily incorporate in the model, but we did not because we wanted to keep the model as simple as possible, and we found it interesting that we get the RT difference anyway, even when we do not introduce asymmetries in the input representations. Indeed, the statistics of the inputs do not dictate in anyway whether the average RT of one group is larger than the other. We studied the different behavioral patterns that spontaneously emerged with different training lengths, and we did not attempt to tune the parameters of the network dynamics to reproduce the actual values of reaction times observed in the experiment but only the observed patterns of reaction times.

We performed the same set of analyzes on the second example RNN, referred to as NET 2, which exhibits a high rule decoding accuracy and reaches the performance criterion with the highest amount of training trials among all the RNNs as shown in Fig. 7B. During cue presentation, this network shows high decoding accuracy for the rule, previous and current response, which are also represented in an abstract format with high CCGP values. However, the shape variable is decoded with much lower accuracy and is not represented in an abstract format (Fig. 8C). Similar to the previous example of NET 1, this network achieves high task performance across different task conditions (Fig. 8D, left). Differently from the previous network example, the reaction time varies based on the rule, irrespective of the shape of the visual cue (Fig. 8D, right). Specifically, it shows a shorter average reaction time for the stay rule compared to the shift rule, regardless of the shape. We applied the same multi-linear regression analyzes to the two example neural networks (Fig. 8E) as the actual data (Fig. 6F). Indeed, we fitted a multi-linear regression model to predict the reaction time on a trial-by-trial basis using three factors: the previous response, the shape of the visual cue, and the rule, and we observed that the rule factor has a stronger weight in predicting reaction times in NET 2 than in NET 1 (Mann–Whitney U test: $p$-value = $10^{-34}$; Fig. 8E). Vice versa, the shape is a stronger factor in predicting the reaction time of NET 1 (Mann–Whitney U test: $p$-value = $10^{-34}$; Fig. 8E). As previously observed in both monkeys, the strongest factor in predicting the reaction time is the interaction of the previous response and the rule in both models because the combination of these two factors is essential for choosing the correct response (see Supplementary Fig. 7A).

To better understand the relation between the representational geometry and reaction time, we analyzed some kinematic properties of the recurrent population activity, from the onset of the visual cue to the response time (Supplementary Fig. 9A). We quantified the trajectory length of population activity in the high dimensional activity space for each visual cue and the mean velocity of the trajectory, whose product is proportional to the average reaction time. In the kinematic space—defined by velocity and trajectory—we introduced two distinct measures: $\delta$(shape) and $\delta$(rule). These measures represent the Euclidean distance between the centroids of conditions based on shape and rule, respectively (Supplementary Fig. 9B). For the two example RNNs, Panel B illustrates the kinematic measures corresponding to each visual cue. We opted to display the logarithm of both the inverse trajectory and velocity because their summation is approximately equal to the logarithm of the average reaction time, as shown by the relationship: log(1/trajectory) + log(velocity) ∼ log(reaction time). Further, we calculated $\delta$(shape) and $\delta$(rule) for each RNN and examined their correlation with the difference in decoding accuracy between shape and rule as measured in the original activity space. The analysis revealed a significant correlation: a higher decoding accuracy for shape in the activity space is associated with a larger $\delta$(shape) in the kinematic space (Supplementary Fig. 9C, left). Conversely, the same is true for the rule condition (Supplementary Fig. 9C, right).

One notable distinction between the model simulations and the actual data is in the representation of the previous response: in both models, it is strongly decoded, whereas in both monkeys, it is not. This discrepancy can be attributed to our simplifying assumption that the previous response variable is entirely disentangled from the visual cue. Moreover, we provided the previous response as an additional input at the cue onset, while in the experiment, animals had to remember it from the previous trial. Another distinction between the models and actual data is the average RTs of the two example networks, which show different values (Fig. 8F). Indeed, again, we did not fit the RTs in the models to reproduce the actual RTs of the animals. Our aim was to reproduce the patterns of RTs for different shapes and rules that emerged naturally from learning. We observed that the reaction time distribution of NET 2 tends to be lower than that of NET 1, and it might be interpretable as a direct consequence of developing a better strategy versus a less memory-efficient (lookup table) strategy. Overall, we observed a huge variability in the average RTs of all the networks (Supplementary Fig. 7B), and it could eventually be rescaled in the model to match the data. In summary, the representational geometries and reaction time patterns observed in the first example RNN (NET 1) resembled the neural and behavioral results observed in Monkey 1 (Figs. 3A, B and 6B, C). This network is probably operating in a lazy regime[30–33], in which the neural representations inherited from the inputs are only slightly modified to perform the task correctly. Notice that shape is disentangled from the other task-relevant variables in the original inputs, and the random projections, despite being non-linear, only partially distort the geometry of the representations (see Supplementary Fig. 8). This explanation is compatible with the observation that this network reaches high performance with more rapid training. Conversely, the results obtained from the second example network (NET 2) mirrored the neural and behavioral outcomes observed in Monkey 2 (Figs. 3C, D and 6D, E). High performance requires longer training (probably a rich regime), which enables the network to learn a representation that better reflects the task structure.

## Discussion

Traditionally, studies on the primate brain focused on the features of the recordings that are conserved across monkeys. It is uncommon to report and discuss differences between monkeys and other animals often because it is difficult to study and interpret them. Here, we showed that it is possible to find clear differences between the representational geometry of two monkeys and that they are associated with subtle but significant behavioral differences. One of the advantages of our approach, based on the analysis of the neural representational geometry, is that it allowed us to study systematically many different interpretable aspects of the geometry of the representation that potentially cause different behaviors. To characterize

the representational geometry, we considered the decoding accuracy and the cross-condition generalization performance (CCGP) for every possible dichotomy of the experimental conditions. The number of dichotomies grows rapidly with the number of conditions, almost exponentially for balanced dichotomies (using Stirling approximation $\sim 2^C / \sqrt{2\pi C}$ where $C$ is the number of conditions). Even though some of the dichotomies are correlated (the full characterization of the geometry requires only $\sim C^2$ numbers), we can still systematically examine a large number of potentially different behaviors. Moreover, the dichotomies are interpretable and often correspond to some of the task key variables. This is the case in our analysis, in which the dichotomies with the highest decoding accuracy and CCGP during the cue presentation correspond to the shape of the visual stimulus for one monkey and the rule for the other. These dichotomies suggested a way to compute the reaction time for different groups of conditions and revealed modest but significant differences in the behavior, which were not detectable from the initial analysis of the bare performance. Although our study is restricted to two monkeys, which is clearly a limit, our methodology can be applied to the systematic analysis of individual differences in an arbitrary number of subjects. Our main result is that it is possible to relate differences in the representational geometry to non-trivial behavioral differences: the way that the conditions are grouped together by the geometry, which is different in the two monkeys, exactly matches the way the conditions are grouped together by the reaction times.

The analysis of the geometry revealed that there is an interesting "structure" in the arrangement of the points that represent different conditions in the firing rate space: for one monkey, the shape of the visual cue is an important variable (a more "visual" monkey), and for the other, it is the rule (a more "cognitive" monkey). This essentially means that for the first monkey, the points corresponding to different conditions in the firing rate space are grouped according to shape if one projects the activity on the coding direction of shape (notice that it is only in this subspace that the points cluster, as in the original space the points are still distinct and allow for the encoding of other variables). Analogously, the points are grouped according to the rule in the other more "cognitive" monkey. Both geometries and even the one in which the points are at random locations in the firing rate space (e.g., when the monkey is basically using a lookup table strategy, for which each visual cue is uniquely associated with a mapping from the previous response to the current response) allow for high performance. This is probably why we cannot see significant differences in the overall performance of the two monkeys. Moreover, the higher value of the shattering dimensionality in Monkey 1 than in Monkey 2 might support the hypothesis of the implementation of a more "visual" strategy to solve the task. However, these geometries have different computational properties that can only be revealed in novel tasks involving generalization or learning of new rules. For example, the more "cognitive" monkey, for which the rule is in an abstract format, would probably learn rapidly a novel task in which the rules are the same but the visual cues change. The new visual cues could be "linked" to the pre-existent groups that represent in an abstract format the two possible rules. The other more "visual" monkey, although at the same learning stage as the "cognitive" monkey, shows stronger representations of the sensory inputs, which, in principle, are useless for performing the task or generalizing to similar new tasks.

The simulation of Recurrent Neural Networks (RNNs) trained to perform the task used in the experiment with high accuracy revealed a significant correlation between the representational geometry of task variables and reaction times and that the two monkeys could have gone through a different amount of training. Indeed, the models showed that those networks that reached the performance criterion with the smallest amount of training also have a higher signal for the shape over the rule, which resembles the representational geometry of Monkey 1. On the other hand, those networks that reached the

threshold with more training developed a stronger signal for the rule than the shape, which resembles the representation in Monkey 2. Unfortunately, we do not have data collected during monkey training, but we know that, due to individual differences in the level of perseveration for which Monkey 2 tended to stay with the same response between consecutive trials, the training process was slightly different. In particular, Monkey 2 required longer training than Monkey 1 to learn to switch between the stay and shift rule and establish the contingency between visual cues and the rule. This empirical note on the behavioral training of the two monkeys is in line with the correlation we observed between the representational geometry of the task variables and the amount of training across all the trained RNNs.

Moreover, with our models, we could reproduce the reaction time patterns, and we studied the relation between the representational geometry of the task variables and reaction time. The models with the highest signal for the shape of the visual cue showed a reaction time that, on average, significantly changes with the shape regardless of the rule, and it resembles the results we observed in Monkey 1. On the contrary, the models with the highest signal for the rule have a reaction time that, on average, significantly changes with the rule regardless of the shape, and it resembles the results we observed in Monkey 2.

It is possible that those networks that reached the performance threshold with a small number of training trials found an optimal policy to solve the task, probably due to the network initialization. Conversely, the RNNs requiring more extensive training developed a reduced representation of the shape, which in principle is useless to solve the task, in favor of a more robust rule representation. The use of different policies as a function of training amount, which is worth investigating in more detail for further studies, might reflect the development of different policies or strategies, also in the two monkeys to solve the same task, mirrored by different representational geometries of task variables correlating with different reaction times. For example, Tsuda et al.[34] recently showed that the different strategies of monkeys and humans in solving a working memory task (monkeys seem to apply a recency-based strategy while humans a target selective strategy[35,36]) could correspond to two different learning stages of a simple recurrent neural network.

Although the model suggests that the differences are due to the training duration, it is also possible that the monkeys would have adopted different policies even at the same learning stage. We know from machine learning studies on curriculum learning that artificial neural networks can solve the same task in different ways depending on the order of presentation of the samples and, more generally, on the details of the learning process[37,38]. Preliminary results on the RNNs indicate that an unbalanced distribution of trial types during the training phase can introduce biases in the resulting policies (data not included). Differences in strategies have been described in experimental studies, in particular in the information representations of the reward[39], in the strategies adopted by two monkeys to solve the same task[40], and in some abstraction tests[41]. A recent study, using a more complex task as the well known pac-man game, has even shown that different strategies can be flexibly switched based on different task demands[42].

In our study, we did not test whether abstract representations could lead to generalization to new stimuli. Introducing a generalization test would have allowed, for example, to test whether the abstract format of the rule in the second monkey generated a faster generalization to a new set of rule cues than in the first monkey. Future studies on abstraction should be planned to test whether the task variables encoded in an abstract form, as opposed to those that are not, would facilitate the generalization of the rules to new items or conditions. The ability of generalization has been reported by several studies on macaques[41,43–45]. For example, Falcone et al.[45] have shown that monkeys can transfer the nonmatch-to-goal rule from the object domain to the spatial domain in a single session, and Sampson et al.[41]

have shown that abstraction can allow generalizing to new conditions, such as new foods, of the rule to choose the worst between two options.

Moving to chronic recordings surely offers the opportunity to follow in time the formation of neural representational geometries by recording before and during the training phases, and after a task is fully learned. Planned behavioral generalization tests to new task conditions are critical to test the relation between the geometry of the representation of a given variable and the animal performance in generalization tasks. These future studies will probably highlight even more individual differences and will allow us to define more precisely what a strategy is and how it is represented in the brain and to predict and test behavioral consequences in a number of novel situations.

## Methods

### Subjects

All the details about the experiment are reported in the original article[22]. Here, we give only a brief description of these details.

Two male rhesus monkeys (Macaca mulatta, 10–11 kg in weight) were trained to perform a visually cued rule-based task. All experimental procedures were in agreement with the Guide for the Care and Use of Laboratory Animals and were approved by the National Institute of Mental Health Animal Care and Use Committee.

Each monkey, while performing the task, sat in a primate chair, with the head fixed in front of a video monitor 32 cm away. An infrared oculometer (Arrington Research, Inc., Scottsdale, AZ) recorded the eye positions.

### Data collection and histology

Up to 16 platinum iridium electrodes (0.5–1.5 MΩ at 1 kHz) were inserted into the cortex with a multielectrode drive (Thomas Recording) to record single-cell activity from dorsolateral prefrontal cortex (Fig. 1C). The recording chambers (18 mm inner diameter) were positioned and angled according to magnetic resonance images (MRI). The single-cell potentials were isolated off-line (Off Line Sorter, Plexon), based on multiple criteria, including principal component analysis, the minimal interspike intervals, and close visual inspection of the entire waveforms for each cell. Eye position was recorded with an infrared oculometer (Arrington Research). The recording sites were localized by histological analysis and MRI (see Tsujimoto et al.[22] for more information).

### The behavioral task

A sequence of the task events of the visually cued rule-based task is shown in Fig. 1A[22,46–48]. For clarity, previous works' authors referred to this task as the visually cued strategy task. The stay and the shift rules were designed as strategies because they represented a simplification of the repeat-stay and change-shift strategies used in previous neurophysiological studies[49,50]. These two strategies were identified by Bussey et al.[51] studying the behavior of monkeys during the learning of visuomotor associations. The monkeys in their study spontaneously adopted the strategies to facilitate learning. As opposed to the previous studies of this task, here we refer to "strategy" as a possible way adopted by the monkey to solve the task, and to "rule" what is instructed to the monkey to perform the task. In each trial, the monkey was required to make a saccade towards one of the two spatial targets, according to a shift or stay rule cued by a visual instruction (Fig. 1B). The appearance of a fixation point (a 0.6° white circle) located at the center of the video screen, with 2 peripheral targets (2.0° white square frames) placed 11.6° to the left and right of the fixation point, represented the beginning of a trial. The monkey had to maintain fixation on the central spot for 1.5 s; after that, a cue period of 0.5 s followed. During the cue period, a visual cue appeared at the fixation point. In each trial, one visual cue was chosen pseudorandomly from a set of four visual cues: a vertical (light gray) or horizontal (light gray) rectangle with the same dimensions (1.0° × 4.9°) and brightness, or a yellow or purple square with the same size (2.0° × 2.0°) (Fig. 1B). Each visual cue instructed either the stay or shift rule. The stay rule, instructed by the vertical rectangle or the yellow square, cued the monkey to choose the same target chosen in the previous trial (as shown in the two consecutive trials' example in Fig. 1A). Conversely, the horizontal rectangle or the purple square instructed the shift rule, which required the monkey to choose the target not chosen in the previous trial. The end of one trial and the beginning of the next one were separated by an intertrial interval of 1 s. The first trial required a random choice of the target since no previous response could be integrated with the information on the current rule. Moreover, in the first trial, the monkey was always rewarded. The monkey had to maintain the fixation on the central point during the whole fixation period (1.5 s) and the cue period (0.5 s) as well as during a subsequent delay period of 1.0, 1.25, or 1.5 s, pseudorandomly selected. The fixation window was a ± 3° square area centered on the fixation point. Both monkeys maintained fixation accurately and rarely made a saccade within the fixation window[22,46]. Any fixation break during the fixation, cue, or delay periods led to abortion of the trial. The fixation point and the two peripheral targets were kept on the screen for the whole duration of the delay period. The disappearance of the fixation spot represented a go signal, instructing the monkey to choose one target by making a saccade to one of them. When the monkey fixated on one of the targets, both squares became filled. The entry of the gaze into the response window was labeled as target acquisition. The monkey had to maintain the fixation on the target for 0.5 s (pre-feedback period). Any fixation break during the pre-feedback period led to abortion of the trial. After the pre-feedback period, in the case of correct response, feedback was provided as a liquid reward (0.2 ml drop of fluid) or, in case of an incorrect response, as red squares over both targets. In the case of an error, the same cue was presented again in the following trial, called the correction trial. Correction trials were presented until the monkey responded correctly. Usually, after an error, there was not more than a correction trial[22,46].

### Neurons and trials sample selection, pseudo-simultaneous population trials, and task conditions definition

We analyzed the neural activity of each monkey separately, only in complete and correct trials, from 400 ms before the cue onset until 500 ms after the cue offset. Linear decoders were trained and tested on pseudo-simultaneous population trials (pseudo trials). We defined a pseudo trial as the combination of spike counts randomly sampled from every neuron in a specific time bin and task condition[2]. The task condition is one of the eight possible combinations of task variables listed in Fig. 1D. We analyzed the activity of neurons recorded in at least five trials per task condition.

Pseudo trials were generated as follows: given a time bin $t$ and task condition $p$, for every neuron, we randomly picked a trial of task condition $p$, and we computed the spike count in the time bin $t$. The single pseudo trial $\gamma$, for condition $p$ at time bin $t$, is then $\gamma^p(t) = (\gamma_1^p(t), \gamma_2^p(t), \ldots, \gamma_N^p(t))$, where $N$ is the number of recorded neurons, and $\gamma_i^p$ ($i$ is the neuron identity, $i = 1, \ldots, N$) is the spike count. We repeated this procedure 100 times, ending up with 100 pseudo trials per task condition and time bin.

Since we did not know a priori which task variables are represented by the neural ensemble, and in order not to introduce any bias in the selection of the task variables to decode, we defined a dichotomy as each pairing of the task conditions in a group of four, for a total of 35 dichotomies[6]. Each dichotomy is a variable that could be decoded. Four of the 35 dichotomies overlap with the task variables. All the other dichotomies cannot be explicitly interpreted in terms of any of the task variables, but rather as a combination of task variables which we referred to as other dichotomies. In particular, the four dichotomies that overlap with the task variables are the previous response,

rule, current response, and the shape of the visual cue (Fig. 1D). The latter identifies whether the visual cue was a rectangle or a square, which could also be interpreted as a gray-colored and non-gray-colored cue.

## Decoding of the neural population activity

For each dichotomy, which is a binary variable, we trained a Support Vector Machine (SVM) classifier with a linear kernel[52] to classify the spike count into either of the two values of the dichotomy. We set a regularization term equal to $10^{-3}$ in all the SVM classifiers. We decoded the neural activity in a 200 ms time bin stepped by 20 ms along time from 400 ms before the cue onset until 500 ms after the cue offset. The linear classifier was trained on pseudo trials built from randomly selected trials. In more detail, for every neuron, we selected 80% of the trials as a training set and the remaining 20% as a testing set to build the pseudo trials. We generated 100 pseudo trials for training and 100 pseudo trials for testing per condition, and this procedure was repeated for 100 iterations. Subsequently, in each iteration, we randomly chose 80% of the training pseudo trials and 20% of the testing pseudo trials to train and test the linear decoder for a total of 100 cross-validations. We showed the final accuracy of the linear decoder as the ratio between the number of correct predictions and the total number of predictions on the testing set averaged across the interactions and cross-validations. To evaluate the statistical significance of the neural signal, we built a null model by randomly shuffling the task condition labels among the pseudo trials. We trained a linear decoder on the shuffled training set for each shuffle, and we assessed its accuracy on the shuffled testing set. We repeated the shuffle procedure 100 times, obtaining a null model distribution. We defined the chance interval as the interval between 2 standard deviations of the null model distribution around the chance level at 50%.

The data were extracted by custom MatLab functions (The MathWorks, Inc., Natick, MA, USA). All decoding analyzes were performed by using scripts of the scikit-learn SVC package along with custom Python scripts[52].

## Neural representation of variables in an abstract format and the Cross Condition Generalization Performance

After assessing which task variables are decoded, we asked in what format they are represented. In particular, we asked whether they are represented in an abstract format. A variable could be defined to be in an abstract format when a linear decoder trained to classify the value of the variable can generalize to new task conditions never used for training. To assess to what extent a variable is in an abstract format, we computed the Cross Condition Generalization Performance (CCGP), that is, the performance of a linear decoder in generalizing to new task conditions not previously used for training[6]. The difference between the traditional cross-validated linear decoder and the cross-condition generalization is in the data used for training and testing the classifier. In the traditional-fashioned decoding analyzes, a decoder is trained on a sub-sample of trials randomly picked from each (experimental) condition, and tested on the held-out trials retained from each condition. In the end, the decoder is trained and tested on all the conditions, and the generalization is only across trials. The CCGP, instead, is computed by training a linear decoder only on a fraction of trials from a subset of conditions and tested on trials belonging to new conditions not used for training. The generalization is now not only across trials but also across conditions.

We assessed the CCGP for each of the 35 dichotomies as follows. Given a dichotomy, defined as a pairing of task conditions in a group of four, we trained the decoder to classify the value of the dichotomy using trials from three task conditions from each side of the dichotomy and tested it on the one held-out condition from each side. Since each side of the dichotomy has four task conditions, there

are 16 possible ways of choosing the training and testing condition set. For each choice of training and testing set, we applied 10 cross-validations, randomly choosing 80% of training trials and 20% of testing trials. We reported the average performance across all 16 possible choices of training and testing conditions and the 10 cross-validations for each dichotomy. To assess the statistical significance of the CCGP, we built a null model where the geometrical structure in the data was destroyed while keeping the variables still decodable[6]. To do that, we applied a discrete rotation to the noise clouds (the trials firing rate of each condition) by permuting the axes of the firing rate space and randomly assigning neural activity to neurons. We repeated this procedure for each cluster separately. We generated 100 null models, and for each of them, we computed the CCGP for all dichotomies, as done on real data. We defined the chance interval for the CCGP measure as the interval between 2 standard deviations of the null model distribution around the chance level at 50%.

## Multi-dimensional scaling analysis

We used the Multi-Dimensional Scaling (MDS) transformation to seek a low-dimensional data representation. We computed the metric MDS, where the dissimilarity matrix was built as follows. We averaged the neural activity in a fixed time bin across pseudo trials within each task condition, and we constructed a $p_c \times p_c$ matrix (with $p_c$ indicating the number of conditions) which stored the Euclidean distance between the average firing rate between each paired condition. To keep information regarding the noise cloud of each task condition, we normalized the Euclidean distance matrix by the squared root of the sum of the variance of each condition along the distance direction between the two clouds. For the analysis based on a single pseudo trial (Fig. 4B), the dissimilarity matrix was defined as a $p_t \times p_t$ matrix, with $p_t$ indicating the total number of pseudo trials across all conditions. This dissimilarity matrix stored the Euclidean distance between the firing rate of each pair of pseudo trials, and it was normalized as described above.

## Behavioral analyzes

We computed the behavioral performance and reaction times of each monkey separately, combining all the sessions we considered for the neural analyzes. We computed the reaction time (RT) only in complete and correct trials. The RT is defined as the time difference between the go signal and target acquisition in each trial. In order not to bias the results due to outliers, we removed those trials with RT larger than 3 standard deviations from the mean. Since the neural analyzes revealed that the difference between the two monkeys comes from different representational geometry of the rule and the shape of the visual cue, we grouped trials per rule (stay-shift) and shape (rectangle-square), for a total of four conditions. We compared the distribution of RTs of trials with different rules and shapes, separately. To test whether the RTs distributions were significantly different, we ran the Mann–Whitney U test ($p$-value < 0.05).

Moreover, we computed the average performance across the sessions for each of the previous four task conditions. The error bar of the estimated average performance was assessed by applying the following formula[53]:

$$\sigma_{+/-} = \frac{Pn + \frac{k^2}{2} \pm k\left[P(1-P)n + \frac{k^2}{4}\right]^{\frac{1}{2}}}{n + k^2}, \tag{1}$$

where $n$ is the number of trials used to compute the performance across sessions, $P$ is the average performance, and $k$ is the confidence level in terms of standard deviation that we fixed equal to 2. We applied the chi-squared test to assess whether the performance was statistically different between different conditions ($p$-value < 0.05).

## Multi-linear regression model for behavior

We fitted a multi-linear regression model on a single-trial basis to better investigate the behavioral differences between the two monkeys. We included only complete and correct trials in the model, and we discarded those trials with reaction times larger than 3 standard deviations from the mean as done in the behavioral analysis. For each trial, we took three independent binary input factors to the model: rule (+1/−1), previous response (+1/−1), and shape (+1/−1). We also included all the interaction terms. The output of the model is the reaction time (RT), and the multi-linear model is defined as follows:

$$
\begin{aligned}
RT = {} & \omega_1 \times [\text{rule}] + \omega_2 \times [\text{previous}] + \omega_3 \times [\text{shape}] \\
& + \omega_4 \times [\text{rule*previous}] + \omega_5 \times [\text{previous*shape}] \\
& + \omega_6 \times [\text{rule*shape}] + \eta,
\end{aligned} \tag{2}
$$

where $\omega_{1,\dots,6}$ are the weights of each factor, and $\eta$ is a constant term. We fitted 100 models, each time randomly subsampling trials from each task condition, in each monkey separately. The number of trials per task condition was set to the minimum number of trials across conditions. We fitted each model by using the ordinary least squares method[54]. We compared the weights' distributions across models between the two monkeys, for each factor, using the Mann–Whitney U test ($p$-value < 0.05).

## Architecture of the Recurrent Neural Network Model

We trained 80 vanilla Recurrent Neural Networks (RNNs) to perform the visually cued rule-based task (see Fig. 7A). Through random fixed weights $W^{rand}$, $N_u = 9$ input units are fully connected to an expansion layer of $M = 100$ rectified linear units (ReLU). The output from the expansion layer is passed through the input weights $W^{in}$ to $N = 100$ recurrent units. $W^{rec}$ defines the recurrent weight matrix ($N \times N$). The readouts of the RNN are a single scalar representing the temporal discounted expected return (value function/critic), and a real vector with a length equal to the total number of possible actions ($N_a = 3$), which are fixation, right or left response (policy/actor). The final action was determined by sampling from the softmax distribution of this vector at each time step.

The input to the network, $\mathbf{u}(t)^{task}$, which is the input vector to the expansion layer, is defined as follows:

$$
\mathbf{u}(t)^{task} = \left( u(t)^{fix}, \mathbf{u}(t)^{shape}, \mathbf{u}(t)^{color}, \mathbf{u}(t)^{prev.resp.} \right), \tag{3}
$$

where $u(t)^{fix}$ is a scalar that is equal to 1 when the network has to fixate, and it is set to 0 when the network is required to provide a response after the delay period; $u(t)^{shape}$ is a one-hot vector of three units encoding the shape of the visual cue (horizontal, vertical rectangle, or square); $u(t)^{color}$ is a one-hot vector of three units encoding the color of the visual cue (gray, yellow, or purple); $u(t)^{prev.resp.}$ is a one-hot vector of two units encoding the previous response (right or left). This is a simplified version of the input with respect to the monkeys' behavioral task where the animals had to retrieve the previous response from the earlier trial (here, we randomly sampled the previous response at the beginning of each trial because we reset the network at the end of each trial for simplicity, and we provided it to the network at the cue onset). For the current model, the input vector $u(t)^{task}$ was randomly selected at the beginning of each trial so that the possible trial types were uniformly randomly sampled.

We defined a positive activity input $u(t)$ to the recurrent units as[55]:

$$
\mathbf{u}(t) = \left[ \mathbf{u}^{(0)} + W^{rand} \mathbf{u}^{task}(t) + \sqrt{2\tau\sigma_{in}^2}\,\boldsymbol{\xi}(t) \right]_+, \tag{4}
$$

where $u^{(0)} = 0.2$ is a constant baseline term (equal for all the units), $W^{rand}$ are random weights from the nine input units $N_u$ to the expansion layer units $M$, $\tau = 100$ ms is the neuronal constant[56], $\sigma_{in} = 0.01$ is the strength

of the input noise, and $\xi(t)$ is Gaussian white noise with zero mean and unit variance, sampled i.i.d. across units and time. Real neurons typically have shorter time constants $\tau$, around 20 ms. In this work, the 100 ms time constant mimics the slower synaptic dynamics based on NMDA receptors[56]. The ReLU non-linearity function $[x]_+ = max(0, x)$ maps the input currents to positive firing rates. The random weights $W^{rand}$ were sampled from a Gaussian with zero mean and standard deviation $1/\sqrt{N_u}$, and they were kept fixed during the whole training phase. We can rewrite the Eq.(4) in the discrete-time description with time step $\Delta t = 20$ ms, using the first-order Euler approximation, in the following way:

$$
\mathbf{u}_t = \left[ \mathbf{u}^{(0)} + W^{rand} \mathbf{u}_t^{task} + \sqrt{\frac{2}{\alpha}\sigma_{in}^2}\, N(\mathbf{0}, \mathbf{I}_M) \right]_+, \tag{5}
$$

where $\alpha = \Delta t/\tau$, and $N(\mathbf{0}, \mathbf{I}_M)$ is the multivariate Gaussian centered in zero with the identity as covariance matrix (of size $M \times M$).

We described the $N$-dimensional recurrent units activity $r(t)$ by the following dynamical equation[57]:

$$
\tau \frac{d\mathbf{r}(t)}{dt} = -\mathbf{r}(t) + \left[ W^{rec}\mathbf{r}(t) + W^{in}\mathbf{u}(t) + \right. \\
\left. + \mathbf{b} + \sqrt{2\tau\sigma_{rec}^2}\,\boldsymbol{\xi}(t) \right]_+, \tag{6}
$$

which, in a discrete-time formulation using first-order Euler approximation, becomes:

$$
\mathbf{r}_t = (1-\alpha)\mathbf{r}_{t-1} + \alpha\left[ W^{rec}\mathbf{r}_{t-1} + W^{in}\mathbf{u}_t + \right. \\
\left. + \mathbf{b} + \sqrt{\frac{2}{\alpha}\sigma_{rec}^2}\, N(\mathbf{0}, \mathbf{I}_N) \right]_+, \tag{7}
$$

where $W^{rec}$ are the recurrent weights, $W^{in}$ are the initial weights from the expansion layer to the recurrent units, b is the bias term, and $\sigma_{rec} = 0.05$ is the strength of the recurrent noise. We initialized the recurrent connection weights $W^{rec}$ as a scaled identity matrix $0.5 \times I_N$, where $I_N$ is the identity matrix of dimension $N \times N$, $N = 100$ recurrent units. The input weights $W^{in}$ connecting the $M = 100$ expansion layer units to the $N = 100$ recurrent units were initialized by sampling them from a Gaussian distribution of mean zero and standard deviation equal to $1/\sqrt{M}$. The bias b term was initialized to zero.

One of the output readouts of the RNN is the scalar representing the temporal discounted expected return $V_t$ (value function/critic) at time $t$, defined as follows:

$$
V_t = W_{critic}^{out}\mathbf{r}_t + b_{critic}. \tag{8}
$$

The second readout that implements the policy (actor) is a real vector of 3 units, each representing a possible action. The final action was determined by sampling from the softmax distribution of this vector as follows:

$$
\mathbf{prob}_t^{actions} = \text{Softmax}\left( W_{actor}^{out}\mathbf{r}_t + \mathbf{b}_{actor} \right). \tag{9}
$$

The output weight matrices $W_{critic}^{out}$ and $W_{actor}^{out}$ were both initialized by sampling from a Gaussian distribution of zero mean and standard deviation equal to $0.4/\sqrt{N}$. The bias terms $b_{critic}$ and $b_{actor}$ were initialized to zero.

For each network, we trained the following parameters: $W^{in}$, $W^{rec}$, $W_{critic}^{out}$, $W_{actor}^{out}$, b, $b_{critic}$ and $b_{actor}$.

## Training of the RNN through the proximal-policy-optimization

We trained the RNNs to perform the visually cued rule-based task using Proximal-Policy-Optimization (PPO), a state-of-the-art deep reinforcement learning algorithm[25,58]. It belongs to the family of policy gradient

algorithms, which optimize the policy directly through gradient ascent, with a focus on maximizing the expected cumulative reward. Other algorithms in this family include REINFORCE and Advantage Actor-Critic (A2C), which have been successfully applied to study neuroscience problems before[27,28]. We defined the Loss Function $\mathcal{L}$ to be maximized on every training batch of trials, as a weighted sum of the PPO policy loss $\mathcal{L}^{PPO}$, the state value function loss $\mathcal{L}^{VF}$, and the entropy regularization term $S$ as follows:

$$\mathcal{L}(\theta) = \mathbf{E}\left[\mathcal{L}_t^{PPO}(\theta) - c_1\mathcal{L}_t^{VF}(\theta) + c_2 S[\pi_\theta](s_t)\right], \quad (10)$$

where $c_1 = 0.5$ and $c_2 = 0.01$ are hyperparameters determining the weights of the value function loss and the entropy regularization term, respectively. E represents the mean over a batch of training trials that is composed by unrolling 20 environments, simulated in parallel with the same agent, for $T = 128$ steps. This batch was then split into 4 mini-batches used for the optimization. $\theta$ refers to the collections of all the trained parameters, and $\pi_\theta$ is the policy used to sample the actions of the network given an input.

The policy loss $\mathcal{L}^{PPO}$ is defined as follows:

$$\mathcal{L}(\theta)^{PPO} = min(\rho_t(\theta)A_t, clip(\rho_t(\theta), 1 - \epsilon, 1 + \epsilon)A_t), \quad (11)$$

where $\rho_t(\theta) = \frac{\pi_\theta(a_t|s_t)}{\pi_\theta^{old}(a_t|s_t)}$ is the probability ratio of the current and old (before the update) policies, $\epsilon$ is a hyperparameter set to 0.1, and $A_t$ is the advantage function (similar to reward-prediction-error) defined as:

$$A_t = -V(s_t) + r_t + \gamma r_{t+1} + \ldots + \gamma^{T-t+1}r_{T-1} + \gamma^{T-t}V(s_T), \quad (12)$$

where the state $s_t = \mathbf{u}_t$ is the input to the network over time as defined in Eq. (5), $r_t$ is the actual reward at time step $t$, $\gamma$ is a standard temporal discount factor set to 0.99, and $V(s_t)$ is the value function computed at the state $s_t$. The goal is to get a policy that maximizes future rewards in the interaction agent/environment loop.

The value function $V(s_t)$ is optimized in a supervised way by minimizing the following mean square error loss:

$$\mathcal{L}^{VF}(\theta) = \frac{1}{2}\left[R_t - V(s_t)\right]^2, \\ R_t = \sum_{i=0}^{k-1}\left[\gamma^i r_{t+1} + \gamma^k V(s_{t+k})\right], \quad (13)$$

where $R_t$ is the $n$-step bootstrapped discounted return at time $t$, $V(s_t)$ is the value function whose output is the expected return from state $s_t$, $\gamma$ is the discounted factor, $k$ is the number of steps until the next state and it is upper bounded by the maximum unroll length $T$, and $[R_t - V(s_t)]$ is the temporal-difference error that provides an estimate of the advantage function for actor-critic.

Finally, the entropy regularization term that helps with exploration over exploitation is defined as follows:

$$S = H(\pi(a_t|s_t)), \quad (14)$$

where $a_t$, and $s_t$, are the action, and the state, respectively, and $H(\pi)$ is the entropy of the policy.

All the parameters were updated via gradient ascent and backpropagation through time using the Adam optimizer with default parameters[59]. In order to deal with the issue of the exploding gradients, we clip the gradient norm always to be ≤1.

## Visually cued rule-based task structure for the network model

We trained the networks to perform the visually cued ruled-based task, which is equivalent to the task that the monkeys were trained on, but without the feedback period (Fig. 1A). The network is reset at the beginning of each trial with initial firing rate $r_0 = 0$. Each trial starts with a fixation period of $t_{fix} = 1500$ ms. The time increment after each step is set to $\Delta t = 20$ ms. During the fixation period, the only non-zero input to the network is the fixation input, and the network has to choose the fixation action to take a reward of 0; otherwise, if it takes a right or left action, the trial is aborted, and a reward of −1 is issued.

After the fixation period, there is the appearance of one of the visual cues (see Fig. 1B), which is randomly sampled at each trial, and it remains on for $t_{cue} = 500$ ms. The fixation input is still active during the cue onset, and the network must continue to maintain fixation. The previous response is provided along with the visual cue. It is randomly sampled in each trial, and it is presented only during the cue period to reproduce the results we showed in Fig. 3, where the previous response is mainly decoded around the visual cue presentation.

After the cue period, a delay period of $t_{delay}$ of 1000, 1200, or 1500 ms is randomly chosen in each trial, as in the monkeys' task[22,46,60], where the only input to the network is the fixation. Again, the network can only choose to maintain fixation, and any other action results in punishment and abortion of the trial.

Subsequently, after the delay period, the fixation input is turned off, representing the go cue, where the network has a maximum of $t_{dec} = 1500$ ms to make a decision, left or right. In this phase, if the network continues to hold fixation for more than $t_{dec}$, the trial is aborted, and a reward of − 1 is issued. If, on the other hand, the network provides the correct action (right or left), a reward of +1 is given, and the trial is terminated. However, if the network makes the wrong choice, it is punished with a reward of −1, and the trial is terminated.

The reaction time (RT) is calculated as the difference between the first time step the network makes a choice, right or left, $t_{fin}$, and the time step of the go cue, $t_{go}$: RT $= t_{fin} - t_{go}$, where $t_{go} = t_{fix} + t_{cue} + t_{delay}$. See Supplementary Fig. 3 for 2 examples of correct trials after successfully training a model.

## Analysis of the correlation between the representational geometry of shape and rule with reaction times in RNNs

Since one main difference in the two monkeys, performing the same task with high accuracy, concerns the representational geometries of the shape of the visual cue and the rule during the cue presentation, we defined two variables, Δ-decoding and Δ-CCGP defined as follows:

$$\begin{aligned}\Delta-\text{decoding} &= \text{accuracy(shape)} - \text{accuracy(rule)}, \\ \Delta-\text{CCGP} &= \text{CCGP(shape)} - \text{CCGP(rule)}, \end{aligned} \quad (15)$$

where accuracy(shape) and accuracy(rule) are the accuracies of the linear decoder in classifying the shape and rule, respectively, during the visual cue presentation, and analogously for the CCGP. We computed these variables for each of the RNNs. They are good indices that briefly summarize the difference in representational geometries of the shape and rule across all the RNNs. Indeed, positive values of Δ-decoding, or Δ-CCGP, suggest a stronger signal of the shape compared to the rule, resembling the representation observed in Monkey 1. Conversely, negative values indicate a stronger signal of the rule compared to the shape, resembling the representation observed in Monkey 2.

The second main result we obtained from neural data is the relation of the representational geometries with reaction times (RTs). To assess the difference in reaction times, we defined a new variable ΔRT as follows:

$$\begin{aligned}\Delta RT &= |\langle \Delta RT(\text{shape})\rangle| - |\langle \Delta RT(\text{rule})\rangle|, \\ \langle \Delta RT(\text{shape})\rangle &= \langle RT(\text{rectangle})\rangle - \langle RT(\text{square})\rangle, \\ \langle \Delta RT(\text{rule})\rangle &= \langle RT(\text{stay})\rangle - \langle RT(\text{shift})\rangle, \end{aligned} \quad (16)$$

where $\langle RT(\text{rectangle})\rangle$, $\langle RT(\text{square})\rangle$, and $\langle RT(\text{stay})\rangle$, $\langle RT(\text{shift})\rangle$, are the average reaction time across trials with rectangle-shaped or square-

shaped cues, and with stay or shift rules, respectively. We then took the absolute value and computed the difference between the two averages.

We assessed the ΔRT for each RNN: positive values of ΔRT indicate that, on average, reaction times are influenced more by the identity of the shape than the rule. Conversely, negative values suggest that reaction times depend more on the rule than on shape.

We subsequently correlated the Δ-decoding, or CCGP, with the number of training trials required to reach the performance threshold of at least 99% of complete trials and 90% of correct trials to stop the training, and with ΔRT, separately.

### Reporting summary

Further information on research design is available in the Nature Portfolio Reporting Summary linked to this article.

## Data availability

The neural data for each subject and the artificial dataset generated in this study are available at the following link: https://github.com/ValeriaFascianelli/geometry-individual-differences.git. The data are also provided in the Supplementary/Source Data file Source data are provided with this paper.

## Code availability

All of our code for this project is written in python, making use of pytorch[61], gym[62], and the broader python scientific computing environment (including numpy[63], scipy[64], matplotlib[65], and scikit-learn[52]). The codes to analyze the neural and artificial datasets are available in the following repository: https://github.com/ValeriaFascianelli/geometry-individual-differences.git.

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

## Acknowledgements

We are grateful to L.F. Abbott for his comments on the manuscript and for insightful discussions. A.G. and S.T. thank S. Wise and A. Mitz for their numerous contributions. We thank M. Alleman, A. Fanthomme, R. Gulli, J. Johnston, J. Minxha, R. Nogueira, L. Posani, K. Rajan, and M. Rigotti for many valuable and knowledgeable discussions. The work was supported by the Simons Foundation, Neuronex (NSF 1707398), the Gatsby Charitable Foundation (GAT3708), the Swartz Foundation, and the Kavli Foundation. A.B. was supported by the Swartz Foundation. A.G. was supported by the Sapienza University of Rome (H2020:PH1181642DB714F6). This work was supported in part through the NYU IT High Performance Computing resources, services, and staff expertise.

## Author contributions

A.G. and S.T. conceived and designed the experiments and collected the data. V.F, F.S., A.G., and S.F. conceptualized and developed the analyzes. V.F. and F.S. analyzed the data under the supervision of S.F. V.F. and A.B. conceived and developed the model under the supervision of S.F. The data was interpreted by V.F., A.B., F.S., S.T., A.G., and S.F., who also wrote the article.

## Competing interests

The authors declare no competing interests.
