## [Peer Review File · Nature Communications]

Neural representational geometries reflect behavioral differences in monkeys and recurrent neural networksREVIEWER COMMENTS

Reviewer #1 (Remarks to the Author):

This is a very interesting paper with interesting ideas. In short the authors have a pretty conventional setup - two monkeys performing a standard rule-following task, while they recorded in DLPFC, a standard area to study rule representation. They find qualitatively different representational geometries in the two monkeys. These geometries correlate with different task strategies.

In any case, the paper is clearly written, cogently argued, and of high quality.

It's also a very unconventional paper. It is, in practice, more like a case study than a standard scientific paper, because there are only two monkeys. Thus certain key claims are, in a technical sense, incorrect. For example, the authors claim that "Here we show that differences between neural representational geometries across subjects correlate with significant differences in their behavior." The term "correlate" here implies a statistically significant correlation. However, correlation cannot be meaningfully performed with a population of two monkeys. The authors are careful not to make such claims with p-values in their Results section, to their credit, but such claims are an integral part of the logic of the paper as a whole.

The authors may be confused here because it is common in monkey studies to use two subjects and treat the number of neurons, rather than the number of monkeys, as the n . This practice is accepted in the monkey community because it is commonly observed that neurons act as independent variables - neurons are not typically correlated with monkey. Thus monkey can be ignored and the second subject is, functionally speaking, an independent replication of the first, rather than a $n=2$. These assumptions are seldom articulated, but they are foundational in the logic of how monkey work proceeds. In the case of the present paper, the situation is different. Here the object of study is a specific representational geometry (in two monkeys) and a specific behavioral strategy (in the same two monkeys). These are variables that refer to the agglomeration of neurons (in the first case) or massed behavior (in the second case). If we blind ourselves to the labels for these, shuffle the labels, the chance we will observe the same pattern by chance again is 25%. In other words, the p-value for the core claim in the paper is, in essence, $p=0.25$.

This does not mean that the results presented here are worthless. Far from it. There is a lot to like about this work. The authors, for example, take great care to make sure that the differences they observe are not due to the particularities of recording site. This is critically important because,

given that there is only an $n=2$, any other potentially confounding variable could explain their results. Most impressively, the authors relate the neural variability to differences in training history.

However, I think that as it stands, this paper doesn't really work because of the unavoidable statistical problem of $n=2$ monkeys.

I have a suggestion for putting the paper on surer footing. I would suggest centering the paper on the RNN results and make them the core of the paper. Here, the ability to perform repeated simulations allows for statistical confidence. These results, while they are "merely" computational/theoretical, are interesting and important enough to form the substance of the paper - they represent an important extension of the (already classic) Bernardi et al. Cell results. Then I would use the monkey data as a secondary set of results, presented as a clearly acknowledged case study that is interesting and consistent with the core claims of the paper. But with those results demoted to a supporting role, their statistical limitations can be more fully acknowledged. They would serve as a kind of proof in principle that the RNN results correspond to things that can be measured in the real world.

In other words, I am arguing that the paper would stand on its own without the physiology results at all. If this were not the case, I would have to recommend rejection. As it is, I think major surgery can save the patient. The physiology results add to the modeling, but only in the sense of enhancing the narrative, not in any probative sense.

Reviewer #2 (Remarks to the Author):

This study characterizes the relation between the reaction times of monkeys performing a rule-based task and the geometry of the state space neural representations of prefrontal activity across different task parameters. The decoding accuracy and CCGP showed that Monkey 1 represents the shape of the visual cue, while for Monkey 2 the rule is the variable with larger decoding power and generalizability. These findings suggest that the first monkey is using a strategy to solve the task that is based on the identity of the individual cue stimuli, while the second used a cognitive strategy based on the rule. These two strategies are accompanied by reaction times that change according with the neural geometry changes. Finally, the RRNs trained to perform the same task showed that the visual cue strategy is associated with networks that reached high performance earlier, while the rule strategy is linked to RNNs that learned more slowly.

This is an interesting paper that is clearly written, has a smooth flow of ideas and contains figures that are clear and well structured. In addition, the study uses a nice set of analytical tools to investigate the representation of tasks variables using neural geometries on a database obtained with proper recording methods on the behaving monkey. The paper provides compelling and novel

information about how the geometry of the activity of neural populations for key behavioral parameters can suggest the use of different strategies to solve the task in individual monkeys. Finally, the learning dynamics of RRNs suggest that visual strategy is present networks that reached high performance earlier and the rule strategy is more prevalent and occurs in networks that learn slowly. A have two main concerns. The first is related to how the prefrontal cortex is integrating the shape or rule information with the previous response to generate the current response. To be able to solve the task the monkeys need information about the previous response and the rule. If the prefrontal activity in Monkey 1 is encoding shape, the output node in the processing circuit must have a strategy based on a lookup table, where each visual cue is uniquely associated with a mapping from the previous response to the current response. Nevertheless, for both strategies the information of the previous response is crucial. Figure 2 shows practically no previous response signal in the two monkeys, while is evident in decoding accuracy - CCGP that there is a steady increase in the current response. A critical question then is how the shape/rule mapping is integrated with the previous response to get the PFC current response coding. The RNNs show a strong previous response in addition to the shape-rule and current response signals, which indicates that all the parameters to solve the task are integrated in the same network. Please provide an explanation.

My second concern is the small changes in reaction times for the different strategy mappings. The difference in the RTs between shape (monkey 1) and rule (monkey 2) is in the order of 5 ms. It is difficult to imagine that these differences reflect different behavioral strategies between animals. Thus, the changes in representational strategy between monkeys, a main conclusion of the paper, is supported by quite slight behavioral differences. Likewise, the differences in reaction times for the RNN are in the order of 0.5 ms (figure 6), which are far from behavioral measurements where the reaction time reflects cognitive load or changes in strategy. In figure 7 the differences of reaction time are around 3 ms in panel B and around 15 ms in panel D, although in the later the reaction times are not realistic, with values in the tens of milliseconds. Furthermore, the average reaction time for the two monkeys is very similar, supporting the notion that both monkeys are at comparable learning stages. Please contrast this fact with the hypothesis that monkey 1 was at an earlier training stage than monkey 2.

Minos comments.

Please provide the recording areas for each monkey.

The paper does not explain why the authors used MDS instead of other dimensional reduction techniques to project the neural activity in a lower dimensional space.

Is the decoding accuracy and CCGP of the RRNs for shape and rule in figure 6 above random?

Reviewer #3 (Remarks to the Author):

In this article, Fascianelli et al. analyse the geometry of the neural representations of task variables in the dorsolateral prefrontal cortex of monkeys, and link two meaningful aspects of the geometry (decoding accuracy and cross-condition generalization performance) to differences in the cognitive strategies used by the monkeys to solve the task. The authors also recapitulate the results in a set of RNNs trained on the same task.

The study is interesting because it introduces a very new and rigorous conceptual framework to (1) link neural activity to subtle aspects of cognition and behavior, thus bridging multiple scales; (2) analyse behavior at the single-subject level and link inter-subject behavioral variability to the neural code; (3) identify different cognitive strategies even when they generate the same behavioral responses; (4) combine different computational approaches (mathematical tools and artificial neural networks) to characterize the geometry of neural representations and to investigate how different geometries and cognitive strategies may emerge during learning; (5) potentially inspire new ways to quantify the relationships between few-shot learning and the formation of abstract representations. For all these reasons, and for the rigor of the analyses, I think that the paper is a very good fit for Nature Communications.

I have a few questions/comments that may help the authors improve the manuscript, in terms of clarifying some interpretations of the results and methodological details:

(1) The decoding accuracy (DA) and cross-condition generalization performance (CCGP) for many of the dichotomies that, in principle, are not task relevant are surprisingly similar to the DA and CCGP of the task relevant dichotomies (Fig.3A). Since the analyses are restricted to correct post-learning trials, I would have expected that most task-irrelevant dichotomies were not decoded with the same accuracy. Can you comment on why that's not the case?

(2) Related to the previous question, I would have expected that the previous response could be decoded in both monkeys, since the interaction between the previous response and the rule is most relevant to learning the task (as also suggested by the multi-linear regression analysis). Would the previous response be decodable with non-linear decoders after denoising the data (similarly to what shown in Fig.3B for shape)? Or is the previous response not decodable because it is maintained in working memory in some other brain areas?

(3) Two other dichotomies (each on half of the conditions) are task relevant: the orientation for the rectangles and the color for the squares. Showing that these dichotomies (on half of the conditions) are encoded in abstract form in Monkey 1 along with the shape would reinforce the interpretation that Monkey 1 implements a lookup table strategy.

(4) In Fig.5F, all the weights are $\sim 10^{-3}$, i.e., three orders of magnitude smaller than the max weight (the interaction term between previous response and rule). If that's the case, what is the significance of such small differences between previous response, rule, and shape in predicting

reaction time? Or is there an error in the normalization? In Supplementary Fig.2, why is the weight of the interaction term different from 1 and only 1 order of magnitude larger than the other weights?

(5) What is the motivation behind using the network architecture shown in Fig.6A, and do you believe the results would be sensitive to the choice of architecture?

(6) What mechanism may give rise to the asymmetries of reaction times in the RNNs, where there aren't asymmetries in the representations of the stimuli?

(7) In Fig.6BC, a bunch of RNNs develop different representational geometries and strategies (different Δ Decoding, Δ CCGP, and Δ RT) but they all do so with the same small (close to 100) number of training trials. Is it the simple stochasticity in the sequence of trial conditions and weight initialization that drives the emergence of different learning effects?

(8) Reading through the paper, I wasn't clear whether the finding of significantly shorter RTs on average for the "rule-based" strategy versus the "lookup-table" strategy was recapitulated in the RNNs. I found the answer at the end, in Supplementary Fig.7, but this is not discussed in the paper. I would advance this result to the main text, since it's interesting and interpretable as a direct consequence of developing a better (abstract) strategy versus a less memory-efficient (lookup table) strategy. The asymmetry of RTs depending on the rule or the shape is also an interesting result, pointing to a specific effect, but it seems more difficult to infer from it a specific difference in the cognitive strategy.

(9) I understand that the analyses on the representational geometry of RNNs are carried out for the recurrent units, except in Supplementary Fig.8, which shows the results for the input expansion layer. I was surprised that in the input layer the representation of the rule is always abstract, but this property is lost in the downstream recurrent units in the RNNs that develop the lookup table strategy. How does this property always emerge upstream and why is it then lost? I thought that abstract representations of the rule would emerge downstream as a result of learning the low dimensional representation of the output.

(10) Very minor:

Line 182: "the cue also differs because the rectangles are grey" → I think it should be "the cue also differs because the rectangles have different orientations", since that's the relevant feature for the task.

Line 622: "the shape and previous response are also represented in an abstract format" → I think it should be "the shape and current response are also represented in an abstract format".

Line 668-671: "One notable distinction between the model simulations and the actual data is in the representation of the previous response: in both models, it is abstract, whereas in both monkeys, it is not" → From Fig.7A: the representation of the previous response in NET1 is not abstract. Related to this, it could be that the previous response is abstract in NET2 because it is provided as an explicit input to the RNN at cue onset, as the authors suggest, but then why is it not abstract in NET1?

Line 1752-1753: "We then computed the difference between the two averages and took the absolute value" → I think you took the absolute values and then computed the difference.

Reviewer #4 (Remarks to the Author):

Review Fascianelli et al, Nat. Comm. Oct 2023

Fascianelli et al apply the representational geometry analysis framework developed in a previous publication from the same group (Bernardi et al, 2020) to investigate neural representations of dorsolateral prefrontal cortex (dlPFC) in nonhuman primates performing a visually-cued rule-based task. They find qualitative differences between the two monkeys, despite comparable average performance, differences that could not be completely accounted for by variability in the location of the electrodes. Training recurrent neural networks (RNN) on the same task reveals similar representational differences across networks, which trace back to differences in the learning process, and manifest in differences in average reaction times (RT). Qualitatively similar RT differences are also found in the animals, suggesting that the across subject coding variability may be due to idiosyncrasies in the training of the two animals.

Novelty and significance: Anyone doing primate work knows that there are often differences across subjects in terms of the strategy they may adopt to solve any given cognitive task, with corresponding differences in the associated neural representations. This is typically addressed by reporting animal by animal breakouts of the statistics being analyzed. It is not often possible to understand such differences in a principled fashion. As such, the attempt at mapping the space of monkey strategies using RNNs is a worthy contribution, especially since it comes with a concrete behavioral prediction that pans out experimentally. Despite these positives, I am not completely convinced by the results as presented: RT distributions are naturally measured when analyzing such behavior and not a natural outcome of typical trained RNNs, so it feels more like a postdiction. The story would be much more compelling if one got more traction out of the modeling in terms of explaining how the identified representational geometries support behavior, and perhaps additional neural predictions about the dynamics of the associated representations.

Major concerns:

Data analysis: Several key steps of the representational geometry analysis are poorly explained and ad hoc. In particular the parametrization of the visual stimulus arbitrarily ignores many features (color, rotation of the shape), despite the fact that the same features are included in the model and critical for defining correct behavior. To make the shape encoding in monkey 1 story compelling, it is important to demonstrate that the qualitative results are robust to changes in this not particularly obvious or justified definition of shape. I find this important because in the absence of the other visual features it is not at all clear why would monkey 1's neural representation suffice to solve the task. Presumably it learns a different policy for all 4 shapes instead of generalizing by rule, but once

the color and orientation are taken out of the equation there is not enough information left to uniquely identify behavioral outcomes.

Modeling: The specific reinforcement learning algorithm used for training the RNN is not widely known, so it is critical to document more precisely how does the network actually end up performing the task. In particular, it would be important to provide a clear explanation for how networks with different representations cause shifts in mean reaction times. It feels the modeling results could be a lot more thorough. Although the RNN is completely observed, it is only analyzed in the same descriptive terms as the animal data, which leaves a significant explanatory gap between neural geometry and behavioral differences.

Writing: The framing of the problem is not as self-contained as one would hope it to be, and the reader is left feeling that reading the Bernardi paper is a prerequisite for understanding what is going on. My suggestion would be to remove some of the repetitive text in the intro that overlaps with first part of results and instead make sure to provide a clear self contained explanation of the key concepts in the results. A graphic depiction would likely help for that. The other main frustration is that it is often hard to understand how the abstract ideas of representational geometry in general are concretely applied to this particular task. More specificity and precision would substantially help make the analysis text more understandable.

Detailed comments/questions:

Data analysis:

- what happens to the rule if there was a mistake in the previous trial? If STAY: is it 'keep doing what would have been the correct action' or 'keep doing what you last did (the incorrect action)'?

can you explain why restrict the analysis to a predefined dataset of 100 pseudo trials? Presumably the variability within this set will restrict variability in cross-validation results (100 folds of a 100 points dataset are not going to be all that different, I'd think fewer folds larger dataset would be a better way of spending the same computational resources). Is there something about the dataset that restricts the number of pseudo trials? In general I would like to see some explanation of this choice.

- CCGP and decoding accuracy are very strongly correlated within animals. Can you explain why? Is it just a matter of accuracy upper bounding CCGP?

- the general framework argues for considering all balanced task variable dichotomies, but it is not clear to me what role do the non-interpretable ones play in the reported results. They are plotted in the figures 2,3 but not mentioned much in the text. What summary statistics actually include all dichotomies? Only the null model? Would it matter if only considering the 4 meaningful ones? Should we even care at all about the other dichotomies?

- What role does nonlinear dimensionality reduction play in the analysis pipeline? I would imagine that it would only be used for visualization in Fig.4 but there is a mention of it in the text in relation to decoding in panel B that I found puzzling since any nonlinear embedding changes the representational geometry.
- Can you comment a little more on why shift is generally faster than stay for both monkeys? This is presumably in general not only after a rule change (?)
- Can one get any traction from analyzing the error trials (maybe in relation to the model errors) or are there too few of those?
- The linear regression analysis of behavior identifies the joint effect of previous action * Rule as the one explaining by far the most variance in both monkeys (Suppl. Fig. 2). Please comment.

Modeling:

- I found the text describing the model weirdly structured, with details of the task and learning procedure intermingled with network architectural description. That paragraph needs cleaning up (starting around line 470 to 500).
- What is the exact value of the discount factor used in the simulations? Am I right in assuming that temporal discounting is the only incentive for shorter RTs (within the allowed range)?
- What is the time constant of the network dynamics and is that compatible to membrane integration time constants? In general, the conversion of the model time axis in ms units could benefit from more explanation.
- The most striking result in Fig 6 B for me is that the shape only strategy is very rare in the networks pool and most networks behave like monkey 2. Can you please comment?
- I found the text describing the differences between classes of networks a little misleading in places, since it is not really about speed or quality of learning per se, but time to reach criterion. May be worth taking a round of editing to the text to make sure that the phrasing is consistent and accurately reflects what's going on in the model.
- Can you comment why the RT for stay is faster than shift in the RNNs (which unless I missed sth is opposite from the two monkeys).

Could one perhaps generate some more refined predictions about RTs in the trial number domain, which could be then tested in the data? - Are RTs in the RNN different somehow in the trial immediate after a rule change for instance?

- Do the RNNs reproduce the multilinear analysis results in the behavior?

Minor:

- shattering dimensionality is mentioned out of the blue in discussion: define/explain or remove (line 771).

- Eq. 7 cuts out of the text width

- Did not understand the logic of analysis described in suppl. fig. 8: isn't the input layer a static map? why would it change at all over learning?

We are grateful to the Reviewers for the insightful comments and suggestions, and we hope we have addressed all the issues satisfactorily. Please, find below our detailed answers to each question enhanced with blue color. We also showed all changes in the revised version of the manuscript with red color.

REVIEWER COMMENTS

Reviewer #1 (Remarks to the Author):

This is a very interesting paper with interesting ideas. In short the authors have a pretty conventional setup - two monkeys performing a standard rule-following task, while they recorded in DLPFC, a standard area to study rule representation. They find qualitatively different representational geometries in the two monkeys. These geometries correlate with different task strategies.

In any case, the paper is clearly written, cogently argued, and of high quality.

It's also a very unconventional paper. It is, in practice, more like a case study than a standard scientific paper, because there are only two monkeys. Thus certain key claims are, in a technical sense, incorrect. For example, the authors claim that "Here we show that differences between neural representational geometries across subjects correlate with significant differences in their behavior." The term "correlate" here implies a statistically significant correlation. However, correlation cannot be meaningfully performed with a population of two monkeys. The authors are careful not to make such claims with p-values in their Results section, to their credit, but such claims are an integral part of the logic of the paper as a whole.

The authors may be confused here because it is common in monkey studies to use two subjects and treat the number of neurons, rather than the number of monkeys, as the n . This practice is accepted in the monkey community because it is commonly observed that neurons act as independent variables - neurons are not typically correlated with monkey. Thus monkey can be ignored and the second subject is, functionally speaking, an independent replication of the first, rather than a $n=2$. These assumptions are seldom articulated, but they are foundational in the logic of how monkey work proceeds. In the case of the present paper, the situation is different. Here the object of study is a specific representational geometry (in two monkeys) and a specific behavioral strategy (in the same two monkeys). These are variables that refer to the agglomeration of neurons (in the first case) or massed behavior (in the second case). If we blind ourselves to the labels for these, shuffle the labels, the chance we will observe the same pattern by chance again is 25%. In other words, the p-value for the core claim in the paper is, in essence, $p=0.25$.

This does not mean that the results presented here are worthless. Far from it. There is a lot to like about this work. The authors, for example, take great care to make sure that the differences

they observe are not due to the particularities of recording site. This is critically important because, given that there is only an $n=2$, any other potentially confounding variable could explain their results. Most impressively, the authors relate the neural variability to differences in training history.

However, I think that as it stands, this paper doesn't really work because of the unavoidable statistical problem of $n=2$ monkeys.

I have a suggestion for putting the paper on surer footing. I would suggest centering the paper on the RNN results and make them the core of the paper. Here, the ability to perform repeated simulations allows for statistical confidence. These results, while they are "merely" computational/theoretical, are interesting and important enough to form the substance of the paper - they represent an important extension of the (already classic) Bernardi et al. Cell results. Then I would use the monkey data as a secondary set of results, presented as a clearly acknowledged case study that is interesting and consistent with the core claims of the paper. But with those results demoted to a supporting role, their statistical limitations can be more fully acknowledged. They would serve as a kind of proof in principle that the RNN results correspond to things that can be measured in the real world.

In other words, I am arguing that the paper would stand on its own without the physiology results at all. If this were not the case, I would have to recommend rejection. As it is, I think major surgery can save the patient. The physiology results add to the modeling, but only in the sense of enhancing the narrative, not in any probative sense.

We are grateful to the Reviewer for the thoughtful comments and suggestions. We agree that the use of the word "correlate" is not justified given that $n=2$. We decided to change the verb 'correlate' (in the title and in many other places) with 'reflect'. What do we mean by that? Why do we believe that the results are interesting even without the model? The quantities that describe the geometries are complex functions of neural activity. One of the advantages of these quantities is that they are interpretable, and hence, not only they allow us to say that the geometries in the two monkeys are different, but also in what they are different. In particular, quantities like CCGP naturally group together some experimental conditions. For example, a high CCGP for shape means that there is one subspace in which the representations of the two gray rectangles cluster together, and they are well separated from the representations of the color squares. When we then considered the behavior, we analyzed 4 reaction times, one for each condition. The fact that the conditions cluster in a way that is similar to the subspace clustering of the neural space is nontrivial: if, for simplicity, we consider only binary reaction times, high or low, then we can have 16 possible patterns for each monkey. This means 256 possible patterns. So, the probability of observing by chance the exact patterns predicted by the analysis of the geometry is pretty small, certainly much smaller than $p=0.25$. This argument is not rigorous in any sense, and it will not be used to compute p values that will be reported in the article. This is only to justify the emphasis we still put on the analysis of the neural data.

We thought of moving the simulations before the description of the experiment, but it would have been difficult to justify the choice of that particular task. As a theoretical study, it could have been done with many other tasks.

So, in the end, we agree with the Reviewer about the misuse of the term 'correlation', and we removed it from the article. We did not completely reorganize the presentation of the results, but we still took seriously all the comments, and we made substantial changes to the text.

First of all, we changed the title of our manuscript to “Neural representational geometries **reflect** behavioral differences in monkeys and recurrent neural networks” in order not to claim any correlation implying statistical significance.

Secondly, we modified the Abstract, Introduction, Results, and Discussion sections in order to present the monkeys' results as an association (or relation) between neural geometry and behavior, which we believe is a more general statement and scientifically more appropriate, as the Reviewer suggested. Nevertheless, we kept describing the RNNs results as a correlation because in these analyses, the correlation is among $n=80$ neural networks, and we assessed the statistical significance using the p -value, which makes the claim of the correlation between representational geometry and behavior in RNNs statistically correct.

Reviewer #2 (Remarks to the Author):

This study characterizes the relation between the reaction times of monkeys performing a rule-based task and the geometry of the state space neural representations of prefrontal activity across different task parameters. The decoding accuracy and CCGP showed that Monkey 1 represents the shape of the visual cue, while for Monkey 2 the rule is the variable with larger decoding power and generalizability. These findings suggest that the first monkey is using a strategy to solve the task that is based on the identity of the individual cue stimuli, while the second used a cognitive strategy based on the rule. These two strategies are accompanied by reaction times that change according with the neural geometry changes. Finally, the RNNs trained to perform the same task showed that the visual cue strategy is associated with networks that reached high performance earlier, while the rule strategy is linked to RNNs that learned more slowly.

This is an interesting paper that is clearly written, has a smooth flow of ideas and contains figures that are clear and well structured. In addition, the study uses a nice set of analytical tools to investigate the representation of tasks variables using neural geometries on a database obtained with proper recording methods on the behaving monkey. The paper provides compelling and novel information about how the geometry of the activity of neural populations for key behavioral parameters can suggest the use of different strategies to solve the task in individual monkeys. Finally, the learning dynamics of RNNs suggest that visual strategy is present networks that reached high performance earlier and the rule strategy is more prevalent and occurs in networks that learn slowly.

I have two main concerns. The first is related to how the prefrontal cortex is integrating the shape or rule information with the previous response to generate the current response. To be able to solve the task the monkeys need information about the previous response and the rule. If the prefrontal activity in Monkey 1 is encoding shape, the output node in the processing circuit must have a strategy based on a lookup table, where each visual cue is uniquely associated with a mapping from the previous response to the current response. Nevertheless, for both strategies the information of the previous response is crucial. Figure 2 shows practically no previous response signal in the two monkeys, while is evident in decoding accuracy - CCGP that there is a steady increase in the current response. A critical question then is how the shape/rule mapping is integrated with the previous response to get the PFC current response coding. The RNNs show a strong previous response in addition to the shape-rule and current response signals, which indicates that all the parameters to solve the task are integrated in the same network. Please provide an explanation.

We completely agree with the Reviewer that the encoding of the previous response is crucial to perform the task. Monkey 1 shows a significant decoding signal (but not CCGP) for the previous response after the stimulus onset, and it remains high for the whole stimulus presentation period (Figure 3A, green line). Interestingly, the previous response signal precedes in time the current response and the rule (Figure 3A, red and blue lines). This is reasonable since after the cue onset, Monkey 1 first encodes the visual shape of the cue, it subsequently recollects the previous response and then encodes the rule instructed by the cue to generate the current response.

Concerning Monkey 2, we agree that there is only a weak previous response signal (Figure 3C, green line). If we look carefully at the recording sites for each monkey separately in Supplementary Figure 1, we found that in Monkey 1, most of the previous response signal comes from the neurons recorded ventrally to the principal sulcus (Supplementary Figure 1B-right). On the other hand, recording sites in Monkey 2 were mainly located dorsally to the principal sulcus (Supplementary Figure 1A-right) where the previous response signal is weak, as in the neurons in Monkey 1. So, we think that the previous response is mostly encoded from neurons placed ventrally to the principal sulcus, but we do not have ventrally recording sites in Monkey 2. However, the main difference between the dorsal and ventral recordings regards only the representation of the previous response, which is encoded mainly in neurons in the ventral sites.

Concerning the RNN simulation, for simplicity, we provided the previous response as a direct input to the network, along with the visual cue randomly chosen for each trial. For this reason, the previous response is strongly encoded because it is entirely disentangled from the other input variables.

We thank again the Reviewer, and we stressed these two important points in the results section of the new version of the manuscript at line 287 and line 766.

My second concern is the small changes in reaction times for the different strategy mappings. The difference in the RTs between shape (monkey 1) and rule (monkey 2) is in the order of 5 ms. It is difficult to imagine that these differences reflect different behavioral strategies between animals. Thus, the changes in representational strategy between monkeys, a main conclusion of the paper, is supported by quite slight behavioral differences. Likewise, the differences in reaction times for the RNN are in the order of 0.5 ms (figure 6), which are far from behavioral measurements where the reaction time reflects cognitive load or changes in strategy. In figure 7 the differences of reaction time are around 3 ms in panel B and around 15 ms in panel D, although in the later the reaction times are not realistic, with values in the tens of milliseconds. Furthermore, the average reaction time for the two monkeys is very similar, supporting the notion that both monkeys are at comparable learning stages. Please contrast this fact with the hypothesis that monkey 1 was at an earlier training stage than monkey 2.

We agree with the Reviewer that the differences in reaction time between shape and rule are small, although it is important to stress that they are statistically significant. Our attention was mainly directed to the different patterns in reaction times rather than the specific values of the reaction times. For this reason, we were interested in studying the different behavioral patterns that spontaneously emerged with different training lengths and we did not attempt to tune the parameters of the network dynamics to reproduce the reaction times observed in the experiment. The idea was to design the simplest neural networks that with a minimal number of assumptions and tuning, could reproduce the patterns of reactions times observed in the experiments. This is definitely a limitation of our study, and we acknowledge that in the new version of the article. Finally, we agree with the Reviewer that the monkeys are at the same learning stage and we added a new sentence in the Discussion of the new version of the manuscript stressing this claim. Our explanation for the representational and the small behavioral differences is that the two monkeys are the at the same learning stage, but it took a different time to reach the same level of performance. We added a paragraph to the new version of the manuscript (lines 621-635 and 689-697).

Minor comments.

Please provide the recording areas for each monkey.

We provided the recording penetration sites for each monkey separately in Supplementary Figure 1.

The paper does not explain why the authors used MDS instead of other dimensional reduction techniques to project the neural activity in a lower dimensional space.

We decided to use the MDS for three reasons: the first reason is that the MDS preserves the pairwise distances between data points. This is crucial to study the representational geometry in terms of the relative distances between points (task conditions) in the firing rate space. Indeed,

MDS ensures that similar neural activity patterns in high-dimensional space remain close to each other in the lower-dimensional representation.

The second reason is that with MDS, it is possible to keep information regarding the noise cloud of each task condition. Indeed, we normalized the Euclidean distance matrix by the squared root of the sum of the variance of each condition along the distance direction between the two clouds (please refer to line 1266 in the Methods section).

The third reason is that it has been previously successfully used by Bernardi et al. (2020) for visualization purposes of representational geometries across many brain areas. We added it to the new version of the manuscript in the Result section, where we described the Multi-Dimensional Scaling plots in detail (lines 372-379).

Is the decoding accuracy and CCGP of the RRNs for shape and rule in figure 6 above random?

The decoding accuracy for shape and rule are both above chance level during the cue presentation, and we reported the value of all the individual decoding accuracies for each RNN in Supplementary Figure 5A (left and right). While for the decoding accuracy, the results are easy to understand, for the CCGP, the results are a bit more complex because not all the RRNs show the shape and the rule in abstract format (Supp Figure 5B).

Reviewer #3 (Remarks to the Author):

In this article, Fascianelli et al. analyse the geometry of the neural representations of task variables in the dorsolateral prefrontal cortex of monkeys, and link two meaningful aspects of the geometry (decoding accuracy and cross-condition generalization performance) to differences in the cognitive strategies used by the monkeys to solve the task. The authors also recapitulate the results in a set of RRNs trained on the same task.

The study is interesting because it introduces a very new and rigorous conceptual framework to (1) link neural activity to subtle aspects of cognition and behavior, thus bridging multiple scales; (2) analyse behavior at the single-subject level and link inter-subject behavioral variability to the neural code; (3) identify different cognitive strategies even when they generate the same behavioral responses; (4) combine different computational approaches (mathematical tools and artificial neural networks) to characterize the geometry of neural representations and to investigate how different geometries and cognitive strategies may emerge during learning; (5) potentially inspire new ways to quantify the relationships between few-shot learning and the formation of abstract representations. For all these reasons, and for the rigor of the analyses, I think that the paper is a very good fit for Nature Communications.

I have a few questions/comments that may help the authors improve the manuscript, in terms of clarifying some interpretations of the results and methodological details:

(1) The decoding accuracy (DA) and cross-condition generalization performance (CCGP) for many of the dichotomies that, in principle, are not task relevant are surprisingly similar to the DA and CCGP of the task relevant dichotomies (Fig.3A). Since the analyses are restricted to correct post-learning trials, I would have expected that most task-irrelevant dichotomies were not decoded with the same accuracy. Can you comment on why that's not the case?

The Reviewer is right that these results require some discussion. The first reason why many dichotomies are decodable is that there are several that are correlated with the task-relevant dichotomies.

There is also a more important second reason why we often observe so many decodable dichotomies. Typically, the representations that we observe in cognitive areas have a relatively low dimensional scaffold that makes CCGP high. However, they're also characterized by significant non-linear distortions, which allow a linear decoder to separate a large number of dichotomies. This is typically a property of high-dimensional representations, and it is observed in pre-frontal cortex and in the hippocampus of non-human primates [Rigotti et al. 2013; Bernardi et al. 2020]. As discussed extensively in Bernardi et al. 2020, these representations are both low-dimensional, as they allow for high CCGP, and high-dimensional, as a linear decoder can separate a large number of different dichotomies. We now added a paragraph to discuss these results and their implications at lines 137-219 and 237-251.

Notice that the situation is different for the CCGP. For a geometry with 8 conditions (and hence 8 points in the activity space), only 3 balanced dichotomies can have a large CCGP. Indeed, the number of dichotomies that are significantly above chance is much smaller than the number of decodable dichotomies. There are still a few dichotomies that are slightly above chance, but they're typically highly correlated to the task-relevant variables.

(2) Related to the previous question, I would have expected that the previous response could be decoded in both monkeys, since the interaction between the previous response and the rule is most relevant to learning the task (as also suggested by the multi-linear regression analysis). Would the previous response be decodable with non-linear decoders after denoising the data (similarly to what shown in Fig.3B for shape)? Or is the previous response not decodable because it is maintained in working memory in some other brain areas?

We thank the Reviewer for asking this question, which was also a major point raised by Reviewer 2. We think the previous response is mostly encoded from neurons placed ventrally to the principal sulcus, as shown in Supplementary Figure 1 for Monkey 1, but unfortunately, we do not have ventrally recording sites in Monkey 2. We want to stress that the main difference between the dorsal and ventral recordings regards only the representation of the previous

response. We agree then with the Reviewer that the previous response is not decodable in Monkey 2 during cue presentation, but it is barely decoded during the fixation period. This can be explained by assuming that the previous response is maintained in working memory ventrally to the principal sulcus. Still, we do not exclude the role of other brain areas as well, like the medial prefrontal cortex, which has been shown to encode the previous response in a cognitive task (Falcone et al., 2017). We discuss this point in the results section at line 287-296.

(3) Two other dichotomies (each on half of the conditions) are task relevant: the orientation for the rectangles and the color for the squares. Showing that these dichotomies (on half of the conditions) are encoded in abstract form in Monkey 1 along with the shape would reinforce the interpretation that Monkey 1 implements a lookup table strategy.

We completely agree with the Reviewer that these dichotomies are task-relevant. However, when we restrict ourselves to the two rectangles or to the two squares, the dichotomies suggested by the Reviewer coincide with the Rule dichotomy. Any significant CCGP (or the lack of it) could be due to the rule signal, which we know is encoded with high accuracy in both monkeys.

(4) In Fig.5F, all the weights are $\sim 10^{-3}$, i.e., three orders of magnitude smaller than the max weight (the interaction term between previous response and rule). If that's the case, what is the significance of such small differences between previous response, rule, and shape in predicting reaction time? Or is there an error in the normalization? In Supplementary Fig.2, why is the weight of the interaction term different from 1 and only 1 order of magnitude larger than the other weights?

We are very grateful to the Reviewer for raising this point. As suggested by the Reviewer, there was a bug in the normalization of the weights in Fig 5F (now it is Fig. 6F) and Supp Fig 2. Indeed, after fixing it, we obtained more meaningful weight values, and the difference between monkeys is clearer. Indeed, the rule has a weight of 3% in monkey 1 vs 12% in monkey 2, while shape is 6% in monkey 1 and $\sim 0\%$ in monkey 2. We normalized the weights to the total sum of the weights. We reported the updated Figure 6F and Supp Fig 2 (as attached below) in the new version of the manuscript, with a correction in the caption of Fig 6F stressing that each weight is normalized to the total sum of the weights.

(5) What is the motivation behind using the network architecture shown in Fig.6A, and do you believe the results would be sensitive to the choice of architecture?

The main reason for using the RNN architecture is to model the reaction times. In our first version of this study, we trained a simple feedforward neural network to perform the task. We found that the representational geometries were similar to those obtained with RNN architectures. In Panel A (see figure below), we report the results from our first model using a two-layer feedforward architecture. The input is passed through two hidden layers with 100 Rectified Linear Units each. We computed the CCGP along the training epochs for the four main task variables averaged across 10 models, and we selected two training periods: Period1 and Period2. Period1 is the set of epochs where the training performance is between 90% and 100%, while Period2 is the range of epochs from 70 to 100 where the training performance is constantly at 100% (Panel B). We observed that in Period1, during the early phase of the high performance period, all dichotomies can be decoded, but only shape is abstract with the highest CCGP (left beeswarm, Panel C). In Period2, all dichotomies can be decoded, but now the rule is in abstract format (right beeswarm, Panel C). Hence, the representational geometry in Period1 resembles the neural representation of Monkey1, where shape is in an abstract format with the highest CCGP, while the representational geometry in Period2 resembles the neural representation of Monkey2, where the rule is in an abstract format. So, the representational geometries observed in a feedforward neural network are compatible with what is observed for an RNN. We also tried not to include the expansion layer in both the RNN and feedforward networks. However, while the overall representational geometries for the task variables remain the same, we observed much less variability among the networks because most of them tend to keep the shape highly decodable throughout the training trials. In conclusion, we observed that the results are not significantly sensitive to the architecture of the network, and the reason we ended up choosing the RNN architecture was to model reaction times.

(6) What mechanism may give rise to the asymmetries of reaction times in the RNNs, where there aren't asymmetries in the representations of the stimuli?

We think that the emerging asymmetries in reaction times are due to the learning of network weights and biases. Indeed, different networks are initialized randomly, and during the training the trials are presented in a random order. Additionally, the learning process is stopped when the performance of the networks is comparable to that of the monkeys (not exactly 100%, but above 90%). This happens before the networks find an optimal policy, which would eliminate any asymmetry in reaction times. These two factors help to explain how different networks can develop behavioral biases, including reaction time asymmetries.

However, to better understand the asymmetries in reaction times in relation to the different representational geometries, we performed a further analyses on the RNNs population activity to answer also Reviewer 4 . We analyzed some kinematic properties of the recurrent population activity, from the onset of the visual cue to the response time (Figure below, Panel A). We quantified the trajectory length of population activity in the high dimensional activity space for each visual cue and the mean velocity of the trajectory, whose product is proportional to the average reaction time. In the kinematic space—defined by velocity and trajectory—we introduced two distinct measures: $\delta(\text{shape})$ and $\delta(\text{rule})$. These measures represent the Euclidean distance between the centroids of conditions based on shape and rule, respectively (refer to Figure, Panel B). For the two example RNNs, Panel B illustrates the kinematic

measures corresponding to each visual cue. We opted to display the logarithm of both the inverse trajectory and velocity because their sum is proportional to the logarithm of the average reaction time, as shown by the relationship: $\log(1/\text{trajectory}) + \log(\text{velocity}) \approx \log(\text{reaction time})$. Further, we calculated $\delta(\text{shape})$ and $\delta(\text{rule})$ for each RNN and examined their correlation with the difference in decoding accuracy between shape and rule as measured in the original activity space. The analysis revealed a significant correlation such that a higher decoding accuracy for shape in the activity space is associated with a larger $\delta(\text{shape})$ in the kinematic space (Panel C, left). Conversely, the same is true for the rule condition (Panel C, right).

We think this additional analysis has provided insights into the relation between representational geometry, assessed in the original activity space, with the kinematics properties of the population activity, which mirrors the asymmetries in reaction time, and we have included it in the revised version of the manuscript. We added this analyses to the new version of the manuscript as Supplementary Figure 9 and a new paragraph in the results section (lines 735-765).

Analysis of the relation between representational geometry and reaction time in RNN. A) Population activity trajectory in the principal component space for the first three axes, showing the activity from cue onset to response for four visual cues for the two example neural networks (Net 1 and Net 2). These trajectories illustrate the representational geometry where, after the cue onset, trajectories with same shapes or rules exhibit closer proximity within the activity space. **B)** The trajectory length and average velocity of the RNN population from cue onset to response define the kinematic space. Here, $\delta(\text{shape})$ and $\delta(\text{rule})$ are the Euclidean distances between the centroids of differing shapes and rules, respectively. Notably, the sum of the

logarithmic transformations of the inverse trajectory length and velocity approximate the logarithm of reaction time, as indicated by the approximate equality: $\log(1/\text{trajectory}) + \log(\text{velocity}) \approx \log(\text{reaction time})$. **C**) Correlation, for all the RNNs, between the distance measures defined in panel B and the difference in the decoding accuracy between shape and rule (assessed in the high dimensional activity space): the higher the decoding accuracy for the shape or rule, the higher the distance in the kinematic space between two different shapes or rules, respectively.

(7) In Fig.6BC, a bunch of RNNs develop different representational geometries and strategies (different $\Delta\text{Decoding}$, ΔCCGP , and ΔRT) but they all do so with the same small (close to 100) number of training trials. Is it the simple stochasticity in the sequence of trial conditions and weight initialization that drives the emergence of different learning effects?

We think that the differences between the networks can be attributed solely to the different initializations and order of presentation of the trials during the training. However, the values on the x-axis of Figures 6B-C (that in the new version of the manuscript is Figures 7B-C) should be multiplied by a factor of 100 (as specified in the x-axis label), so the approximately 100 trials the reviewer is referring to are actually about 10000 trials.

(8) Reading through the paper, I wasn't clear whether the finding of significantly shorter RTs on average for the "rule-based" strategy versus the "lookup-table" strategy was recapitulated in the RNNs. I found the answer at the end, in Supplementary Fig.7, but this is not discussed in the paper. I would advance this result to the main text, since it's interesting and interpretable as a direct consequence of developing a better (abstract) strategy versus a less memory-efficient (lookup table) strategy. The asymmetry of RTs depending on the rule or the shape is also an interesting result, pointing to a specific effect, but it seems more difficult to infer from it a specific difference in the cognitive strategy.

We followed the suggestion of the Reviewer, and we moved Supplementary Figure 7 to the main text (Figure 8F). We discussed these results in the new version of the manuscript (lines 776-790).

(9) I understand that the analyses on the representational geometry of RNNs are carried out for the recurrent units, except in Supplementary Fig.8, which shows the results for the input expansion layer. I was surprised that in the input layer the representation of the rule is always abstract, but this property is lost in the downstream recurrent units in the RNNs that develop the lookup table strategy. How does this property always emerge upstream and why is it then lost? I thought that abstract representations of the rule

would emerge downstream as a result of learning the low dimensional representation of the output.

We thank the Reviewer for the question, as we realized that Supplementary Figure 8 required further explanation. We would like to clarify that the representation of the rule and shape in the input layer, as in Supplementary Figure 8, is not shown as a function of learning. Rather, it shows the representation of variables for each RNN at their different learning stage, defined by the number of training trials required to reach a performance threshold of 90%. Therefore, the representations shown in the expansion layer are after learning. The results suggest that when the networks achieve high performance, the rule and shape are represented with high accuracy and CCGP in the expansion layer of all networks. However, in some networks, they are lost in the recurrent units due to the varying amount of training trials required to reach the same learning stage. We appreciate the Reviewer for raising this important issue, and we have modified the caption of Supplementary Figure 8 in the new version of the manuscript accordingly.

(10) Very minor:

Line 182: "the cue also differs because the rectangles are grey" → I think it should be "the cue also differs because the rectangles have different orientations", since that's the relevant feature for the task.

We changed the sentence in the text as the Reviewer suggested.

Line 622: "the shape and previous response are also represented in an abstract format" → I think it should be "the shape and current response are also represented in an abstract format".

We corrected this mistake in the new version of the manuscript. We thank the Reviewer for noticing it.

Line 668-671: "One notable distinction between the model simulations and the actual data is in the representation of the previous response: in both models, it is abstract, whereas in both monkeys, it is not" → From Fig.7A: the representation of the previous response in NET1 is not abstract. Related to this, it could be that the previous response is abstract in NET2 because it is provided as an explicit input to the RNN at cue onset, as the authors suggest, but then why is it not abstract in NET1?

We thank the Reviewer for pinpointing it. Indeed, the Reviewer is right and we changed the text specifying that the difference is mainly in the decoding accuracy of the previous response rather than in the CCGP. Since the previous response is provided as input to the network, it

guarantees that it is significantly decoded in both networks; however, the reason why it is not abstract in NET2 is not probably due to the input structure but rather to the regime that the network develops during the training to solve the task. We commented on this in the new version of the manuscript at lines 766-775.

Line 1752-1753: "We then computed the difference between the two averages and took the absolute value" → I think you took the absolute values and then computed the difference.

The Reviewer is totally right, and we corrected the mistake in the new version of the manuscript (line 1572 in the new version of the manuscript).

Reviewer #4 (Remarks to the Author):

Review Fascianelli et al, Nat. Comm. Oct 2023

Fascianelli et al apply the representational geometry analysis framework developed in a previous publication from the same group (Bernardi et al, 2020) to investigate neural representations of dorsolateral prefrontal cortex (dlPFC) in nonhuman primates performing a visually-cued rule-based task. They find qualitative differences between the two monkeys, despite comparable average performance, differences that could not be completely accounted for by variability in the location of the electrodes. Training recurrent neural networks (RNN) on the same task reveals similar representational differences across networks, which trace back to differences in the learning process, and manifest in differences in average reaction times (RT). Qualitatively similar RT differences are also found in the animals, suggesting that the across subject coding variability may be due to idiosyncrasies in the training of the two animals.

Novelty and significance: Anyone doing primate work knows that there are often differences across subjects in terms of the strategy they may adopt to solve any given cognitive task, with corresponding differences in the associated neural representations. This is typically addressed by reporting animal by animal breakouts of the statistics being analyzed. It is not often possible to understand such differences in a principled fashion. As such, the attempt at mapping the space of monkey strategies using RNNs is a worthy contribution, especially since it comes with a concrete behavioral prediction that pans out experimentally. Despite these positives, I am not completely convinced by the results as presented: RT distributions are naturally measured when analyzing such behavior and not a natural outcome of typical trained RNNs, so it feels more like a postdiction. The story would be much more compelling if one got more traction out of the modeling in terms of explaining how the identified representational geometries support behavior, and perhaps additional neural predictions about the dynamics of the associated representations.

Major concerns:

Data analysis: Several key steps of the representational geometry analysis are poorly explained and ad hoc. In particular the parametrization of the visual stimulus arbitrarily

ignores many features (color, rotation of the shape), despite the fact that the same features are included in the model and critical for defining correct behavior. To make the shape encoding in monkey 1 story compelling, it is important to demonstrate that the qualitative results are robust to changes in this not particularly obvious or justified definition of shape. I find this important because in the absence of the other visual features it is not at all clear why would monkey 1's neural representation suffice to solve the task. Presumably it learns a different policy for all 4 shapes instead of generalizing by rule, but once the color and orientation are taken out of the equation there is not enough information left to uniquely identify behavioral outcomes.

We thank the Reviewer for stressing this point, which has also been raised by Reviewer 3. We agree that the decoding and CCGP of other visual features would strengthen the results, particularly concerning the hypothesis of implementing a lookup table strategy in Monkey 1. We thought about studying the other visual features like the rotation of the rectangles (vertical vs horizontal) and the colors of the squares (yellow vs purple), but we realized that there is a profound conceptual problem. Indeed, there is a 100% correlation of rotation and color features with the rule itself, making any further claim very weak. Indeed, by construction, different orientations of the rectangles instruct different rules, and different square colors instruct different rules. Any significant accuracy we might see could be due to the rule signal, which we know is encoded with high accuracy in both monkeys. For all these reasons, we decided to label one of the dichotomies as a visual feature (the shape) that is intrinsically not correlated with any of the other variables. This dichotomy is not defined a priori, it is just one of the dichotomies, and it happens to be the one with the highest decoding accuracy and CCGP in one of the monkeys.

Modeling: The specific reinforcement learning algorithm used for training the RNN is not widely known, so it is critical to document more precisely how does the network actually end up performing the task. In particular, it would be important to provide a clear explanation for how networks with different representations cause shifts in mean reaction times. It feels the modeling results could be a lot more thorough. Although the RNN is completely observed, it is only analyzed in the same descriptive terms as the animal data, which leaves a significant explanatory gap between neural geometry and behavioral differences.

The Proximal Policy Optimization (PPO) algorithm used in this paper is a deep reinforcement learning algorithm that belongs to the family of policy gradient algorithms, which optimize the policy directly through gradient ascent, with a focus on maximizing the expected cumulative reward. Other algorithms in this family include REINFORCE and Advantage Actor-Critic (A2C), which have been successfully applied to study neuroscience problems before. For instance, Song HF, Yang GR, Wang X-J (2017) used reward-based training of recurrent neural networks for cognitive and value-based tasks, while Wang JX, Kurth-Nelson Z, Kumaran D, Tirumala D, Soyer H, Leibo JZ, Hassabis D, Botvinick M. (2018) studied the prefrontal cortex as a meta-reinforcement learning system. Although there may be differences in the convergence of these algorithms, the idea behind them is the same, and the results do not depend on the specific choice of algorithm type, as long as we stay within the domain of policy gradient

algorithms. We thank the Reviewer for this suggestion and we added a better explanation of PPO algorithm in the Method section at lines 1434-1441.

We appreciate the Reviewer suggestion to further explore the relation between representational geometry and reaction time. For this reason, we analyzed some kinematic properties of the recurrent population activity, from the onset of the visual cue to the response time (see figure below, Panel A). We quantified the trajectory length of population activity in the high dimensional activity space for each visual cue and the mean velocity of the trajectory, whose product is proportional to the average reaction time. In the kinematic space—defined by velocity and trajectory—we introduced two distinct measures: $\delta(\text{shape})$ and $\delta(\text{rule})$. These measures represent the Euclidean distance between the centroids of conditions based on shape and rule, respectively (refer to Figure, Panel B). For the two example RNNs, Panel B illustrates the kinematic measures corresponding to each visual cue. We opted to display the logarithm of both the inverse trajectory and velocity because their summation is approximately equal to the logarithm of the average reaction time, as shown by the relationship: $\log(1/\text{trajectory}) + \log(\text{velocity}) \approx \log(\text{reaction time})$. Further, we calculated $\delta(\text{shape})$ and $\delta(\text{rule})$ for each RNN and examined their correlation with the difference in decoding accuracy between shape and rule as measured in the original activity space. The analysis revealed a significant correlation such that a higher decoding accuracy for shape in the activity space is associated with a larger $\delta(\text{shape})$ in the kinematic space (Panel C, left). Conversely, the same is true for the rule condition (Panel C, right).

We think this additional analysis has provided insights into the relation between representational geometry, assessed in the original activity space, with the kinematics properties of the population activity, which mirrors the average reaction time, and we have included it in the revised version of the manuscript.

Analysis of the relation between representational geometry and reaction time in RNN. A)

Population activity trajectory in the principal component space for the first three axes, showing the activity from cue onset to response for four visual cues for the two example neural networks (Net 1 and Net 2). These trajectories illustrate the representational geometry where, after the cue onset, trajectories with same shapes or rules exhibit closer proximity within the activity space. **B)** The trajectory length and average velocity of the RNN population from cue onset to response define the kinematic space. Here, $\delta(\text{shape})$ and $\delta(\text{rule})$ are the Euclidean distances between the centroids of differing shapes and rules, respectively. Notably, the sum of the logarithmic transformations of the inverse trajectory length and velocity approximate the

logarithm of reaction time, as indicated by the approximate equality: $\log(1/\text{trajectory}) + \log(\text{velocity}) \approx \log(\text{reaction time})$. **C)** Correlation, for all the RNNs, between the distance measures defined in panel B and the difference in the decoding accuracy between shape and rule (assessed in the high dimensional activity space): the higher the decoding accuracy for the shape or rule, the higher the distance in the kinematic space between two different shapes or rules, respectively.

Writing: The framing of the problem is not as self-contained as one would hope it to be, and the reader is left feeling that reading the Bernardi paper is a prerequisite for understanding what is going on. My suggestion would be to remove some of the repetitive text in the intro that overlaps with first part of results and instead make sure to provide a clear self contained explanation of the key concepts in the results. A graphic depiction would likely help for that. The other main frustration is that it is often hard to understand how the abstract ideas of representational geometry in general are concretely applied to this particular task. More specificity and precision would substantially help make the analysis text more understandable.

We thank the Reviewer, and we tried to be clearer in writing the result section to make the understanding of the results self-consistent without relying on any previously published paper. We removed some of the repetitive text from the introduction and we added a new paragraph in the results section (lines 137-219) containing an explanation of the key concepts analyzed in the paper. Moreover, we added a schematic in the new version of the manuscript (Figure 2) to better understand the following results. We report below the schematic we added to the main text as Figure 2.

Schematic of different representational geometries for 4 conditions in the neural activity space and their properties. A) Left: factorized or disentangled representations where the 4 points are arranged on a square. The shape (circle vs triangle) and color (red vs blue) are encoded along two orthogonal directions. This geometry allows the representation of shape (and color) in abstract format, i.e. high CCGP. Right: Random representation where the 4 points are placed at random locations in the activity space. This geometry does not support the representation of the shape in abstract format, i.e. low CCGP. **B)** Left: Low shattering dimensionality, where the 4 points are placed at the vertices of a square. The shattering dimensionality is low because not all the dichotomies can be decoded by a linear decoder due to the XOR configuration (purple and green circles). Right: High shattering dimensionality supports the decoding of a higher number of dichotomies, including the one not linearly decodable, i.e. XOR, in a low dimensional representation.

Detailed comments/questions:

Data analysis:

- what happens to the rule if there was a mistake in the previous trial? If STAY: is it 'keep doing what would have been the correct action' or 'keep doing what you last did (the incorrect action)'?

We are grateful to the Reviewer for asking this question as we did not describe all the details of what happens after a mistake. Indeed, in the experiment, every error trial was followed by a correction trial, where the same rule was instructed, and the monkey had to respond with what would have been the correct action. After errors, the monkeys performed correction trials almost perfectly, requiring no more than one correction trial per session. We added these details to the new version of the manuscript both in the method section at line 1109 and result section at line 230.

- can you explain why restrict the analysis to a predefined dataset of 100 pseudo trials? Presumably the variability within this set will restrict variability in cross-validation results (100 folds of a 100 points dataset are not going to be all that different, I'd think fewer folds larger dataset would be a better way of spending the same computational resources). Is there something about the dataset that restricts the number of pseudo trials? In general I would like to see some explanation of this choice.

We agree that 100 folds of a 100 points dataset would not be ideal, and indeed it is not what we did, and we are sorry that it wasn't clear. In each of 100 iterations, we selected 80% and 20% of the trials from each single neuron for the training and testing set, respectively. With these trials, we generated 100 pseudo trials/condition for the training set and 100 pseudo trials/condition for the testing set. Therefore, in each iteration, we generated two sets of 100 pseudo trials: 100 pseudo trials for training and 100 pseudo trials for testing. For each set of pseudo trials, we ran 100 cross-validations. In each cross-validation, we randomly select the 80% of training pseudo trials and 20% of testing pseudo trials to train and test a linear decoder. In total, we generated 100 pseudo trials/condition (100 training+100 testing) for 100 times, and for each pseudo trial set realization, we ran 100 cross-validations, for a total of 100x100 validations.

We updated the Method section of the new version of the manuscript to better specify the generation of pseudo trials and the cross-validation procedure (lines 1172-1178).

- CCGP and decoding accuracy are very strongly correlated within animals. Can you explain why? Is it just a matter of accuracy upper bounding CCGP?

CCGP and decoding accuracy (DA) are often correlated because CCGP typically cannot be larger than DA. CCGP is some form of out-of-distribution (OOD) generalization, and unless one constructs a rather peculiar situation, it is always smaller than DA. This is not surprising: if a

variable is not encoded (DA at chance), there is no way that CCGP is large. However, there are several situations in which they are not correlated (e.g. in Bernardi et al. 2020, context is strongly encoded (DA is almost at 100%), but CCGP is at chance). In our case, the CCGP and DA of the most relevant variables are indeed correlated. However, some variables, like the previous response (green line) and rule (blue line) in Monkey 1, are significantly decoded during the cue presentation (Figure 3A), but their CCGP is not significantly different from chance (Figure 3B).

- the general framework argues for considering all balanced task variable dichotomies, but it is not clear to me what role do the non-interpretable ones play in the reported results. They are plotted in the figures 2,3 but not mentioned much in the text. What summary statistics actually include all dichotomies? Only the null model? Would it matter if only considering the 4 meaningful ones? Should we even care at all about the other dichotomies?

We typically show the results for all the dichotomies (interpretable and not interpretable) because we want to have an unbiased approach to the analysis of all possible variables. This might help us to discover unexpected variables that are represented in an abstract format (see Courellis, H.S., Mixha, J., Cardenas, A.R., Kimmel, D., Reed, C.M., Valiante, T.A., Salzman, C.D., Mamelak, A.N., Fusi, S. and Rutishauser, U., 2023. Abstract representations emerge in human hippocampal neurons during inference behavior. bioRxiv, pp.2023-11 for an interesting example: the variable corresponding to the dichotomy that groups together stimuli with the same responses turned out to be unexpectedly one of the variables with the largest CCGP). In general, it is not guaranteed that the variables in an abstract format correspond to the most relevant task variables. It is particularly important to look at all dichotomies when focusing on individual differences.

Moreover, in terms of summary statistics, as the Reviewer correctly suggested, the non-interpretable dichotomies play a role in the null model along with the main 4 variables. We agree that, in general, for the message of the paper, the results from the non-interpretable dichotomies are not essential, but they convey interesting information about the overall statistics of the decoding accuracy and CCGP. For example, all the non-interpretable dichotomies contribute to the shattering dimensionality (as explained in Figure 2 of the new version of the manuscript).

- What role does nonlinear dimensionality reduction play in the analysis pipeline? I would imagine that it would only be used for visualization in Fig.4 but there is a mention of it in the text in relation to decoding in panel B that I found puzzling since any nonlinear embedding changes the representational geometry.

The dimensionality reduction we applied to visualize the data reported in Figure 4 (Figure 5 in the new version of the manuscript) is a linear multi-dimensional scaling based on the Euclidean distance. We agree with the Reviewer that non-linear dimensionality reduction methods would

distort the representational geometry. We added a few sentences justifying the use of a multidimensional scaling plot at lines 372-379 in the new version of the manuscript and we specified that it's linear.

- Can you comment a little more on why shift is generally faster than stay for both monkeys? This is presumably in general not only after a rule change (?)

The Reviewer is right that Monkey 2 shows average reaction times which are faster for shift than stay trials. Nevertheless, we didn't find any significant difference in Monkey 1 (Figure 6C-E). A previous study on monkeys performing a task where the animal has to maintain or shift the choice according to a strategy did not observe any significant difference between stay or shift (Genovesio, Brasted, Mitz, and Wise, 2005, Supplementary Table 3). Moreover, in our previous study (Fascianelli et al., 2020) we analyzed the reaction times of the monkeys after a rule change, and we did not find any significant difference between repeating and shifting rules in consecutive trials. This isolated case of faster reaction times in shift trials in one monkey makes it hard to propose a general explanation beyond the single case. Moreover, in the RNNs studies, we found both situations where networks have "stay" faster than "shift" and vice versa, so we did not find any significant imbalance between stay and shift.

- Can one get any traction from analyzing the error trials (maybe in relation to the model errors) or are there too few of those?

Unfortunately, there are too few error trials per session, and it is not guaranteed to have enough trials in each task condition to perform meaningful decoding (and CCGP) analyses.

- The linear regression analysis of behavior identifies the joint effect of previous action * Rule as the one explaining by far the most variance in both monkeys (Suppl. Fig. 2). Please comment.

The strongest factor in predicting the reaction time is the interaction of the previous response and the rule in both monkeys. Combining these two factors is essential for choosing the correct response. Indeed, both monkeys have been trained to combine the previous response and rule to provide the current correct response. We added this comment to the new version of the manuscript in the result section at lines 452-459.

Modeling:

- I found the text describing the model weirdly structured, with details of the task and learning procedure intermingled with network architectural description. That paragraph needs cleaning up (starting around line 470 to 500).

We thank the Reviewer for the suggestion. We cleaned up the paragraph, and now we think that the narrative is more linear. In particular, we moved the description of the architecture to the very beginning of the paragraph, followed by details of the task and learning procedure (line 1335-1350).

- What is the exact value of the discount factor used in the simulations? Am I right in assuming that temporal discounting is the only incentive for shorter RTs (within the allowed range)?

We used the discount factor with a value of 0.99 (a standard value) in all simulations. We have specified this value at line 1464 of the methods section. In general, in reinforcement learning, the discount factor plays a crucial role in determining how much the agent prioritizes immediate rewards over future rewards, thus influencing its decision-making strategy. In our context, where there are no specific task constraints that require networks to find solutions with shorter reaction times (such as providing a reward inversely proportional to the reaction time), the discount factor is the only factor that influences the average reaction time.

- What is the time constant of the network dynamics and is that compatible to membrane integration time constants? In general, the conversion of the model time axis in ms units could benefit from more explanation.

We set the neuronal time constant τ equal to 100 ms in all the simulations, and we specified it in the method section. Real neurons typically have shorter time constants, around 20 ms. However, the time constant of individual neurons is not the relevant time scale when the dynamics of a network of spiking neurons is investigated. Typically, the dynamics is dominated by the time constants of synaptic transmission [Brunel, Nicolas. "Dynamics of sparsely connected networks of excitatory and inhibitory spiking neurons." *Journal of computational neuroscience* 8 (2000): 183-208]. In our work, the 100 ms time constant mimics the slower synaptic dynamics based on NMDA receptors [see also Wang, X.-J. Probabilistic decision making by slow reverberation in cortical circuits. *Neuron* 36, 955–968 (2002)]. We added a paragraph to the new version of the manuscript (methods, lines 1382-1385).

- The most striking result in Fig 6 B for me is that the shape only strategy is very rare in the networks pool and most networks behave like monkey 2. Can you please comment?

We believe that the shape of the visual input is not an essential and unique feature to elaborate the correct response. Nevertheless, since the shape information is provided in the input, it is decoded in the expansion layer of all networks with high accuracy. However, most networks will not decode the shape at the end of the training. Indeed, the behavioral task aimed to recognize the rule indicated by the visual cue. For this reason, most networks developed representations more similar to those of Monkey 2, which we could consider as the more "natural" way of solving the task, by associating the rule with the previous response rather than relying on the shape as a relevant feature.

- I found the text describing the differences between classes of networks a little misleading in places, since it is not really about speed or quality of learning per se, but time to reach criterion. May be worth taking a round of editing to the text to make sure that the phrasing is consistent and accurately reflects what's going on in the model.

The Reviewer is correct in pointing out that the differences among networks are not related to the quality of learning per se. In fact, we stop training the network as soon as it reaches a performance of 90%, so all networks are at the same learning stage. However, the number of training trials to get there is different for different networks. We described the networks along a continuum spectrum of training lengths, ranging from those that reached the learning stage with fewer training trials, to those that required a higher number of training trials to reach the same performance threshold. We did not intend to define classes of networks that assume a discrete classification of the networks according to training lengths or learning stages. We edited the text in the Results section in the new version of the manuscript in order to be clearer about the differences among networks (lines 558-562).

- Can you comment why the RT for stay is faster than shift in the RNNs (which unless I missed sth is opposite from the two monkeys). Could one perhaps generate some more refined predictions about RTs in the trial number domain, which could be then tested in the data?

The claim that the reaction time for "stay" is faster than for "switch" is only valid for the given network example. Generally, the training process leads to differences in RT in both directions. So we have networks for which the RT for stay is faster than shift, and other networks for which RT for shift is faster than stay. What tends to be preserved is the grouping of the conditions: the two conditions of stay have approximately the same RT, and the two conditions of shift exhibit a different RT. The important thing to note is that due to the random initialization of parameters and the random sequence of trials during learning, these asymmetries naturally develop in the networks. We thank the Reviewer and we added a paragraph in the results section of the new version of the manuscript at lines 621-635, 689-697.

- Are RTs in the RNN different somehow in the trial immediate after a rule change for instance?

Since there is no temporal relation between consecutive trials in the RNNs, it is not possible to estimate if there is a change in the RT after the rule change. This is because, at the beginning of each trial, the network is reset with an initial firing rate equal to zero, so there is no "memory" of the rule of the previous trial to define a rule change (the previous response, which is needed to perform the task, is assumed to be preserved across trials in a different area and it is provided as an additional input to the RNN that we simulated). We stressed this point in the method

section at line 1495. Nevertheless, we have previously analyzed from actual data if the reaction time significantly changed between the switch and non-switch rule (consecutive trials with the same and different rule), and we did not find any significant differences in both monkeys (Fascianelli et al., 2020).

- Do the RNNs reproduce the multilinear analysis results in the behavior?

We thank the Reviewer for this question, and we applied the same multi-linear regression analyses to the neural networks as the actual data. We observed that the two example RNNs reproduced behavioral results which are compatible with those observed in the two monkeys. We are grateful to the Reviewer for suggesting extending this analysis to the model, and we included the new results shown below in the new version of the manuscript as Figure 8E and Supplementary Figure 7A.

Minor:

- shattering dimensionality is mentioned out of the blue in discussion: define/explain or remove (line 771).

We added a schematic explaining the shattering dimensionality at the very beginning of the results in the new version of the manuscript (see Figure 2). We inserted new paragraphs in the results section (lines 137-219, 237-251) defining the shattering dimensionality, with particular attention to the results discussed in this manuscript.

- Eq. 7 cuts out of the text width

We formatted both Eq. 6 and 7 to fit the text width.

- Did not understand the logic of analysis described in suppl. fig. 8: isn't the input layer a static map? why would it change at all over learning?

The Reviewer is right that the input layer is a static map, and the variable representation is not expected to change over learning. We thank the Reviewer for pinpointing this issue, which was also raised by Reviewer 3, as we realized that Supplementary Figure 8 required further explanations. We would like to clarify that the representation of the rule and shape in the input layer, as in Supplementary Figure 8, is not shown as a function of learning. Rather, it shows the representation of variables for each RNN at their different learning stage, defined by the number of training trials required to reach the performance threshold of 90%. Therefore, the representation in the expansion layer is after learning, and there is not any learning component. The results suggest that when the networks have achieved high performance, the rule and shape are represented with high accuracy and CCGP in the expansion layer of all networks. However, in some networks, they are lost in the recurrent units due to the varying amount of training trials required to reach the same learning stage (90%). We appreciate the Reviewer for raising this important issue, and we have modified the caption of Supplementary Figure 8 in the new version of the manuscript accordingly.

REVIEWERS' COMMENTS

Reviewer #1 (Remarks to the Author):

The authors have addressed my concerns in a way that I think is satisfactory.

Reviewer #2 (Remarks to the Author):

The authors did a nice job answering all the reviewers' comments. However, in line with reviewer 1 the paper needs in the discussion a precautionary note regarding the (over)interpretation of distinct neural geometries in only two monkeys with differences in reaction times of 5 ms.

Reviewer #3 (Remarks to the Author):

The authors have provided thorough responses to all my questions. I have read the responses to the other reviewers (some of which overlapped with my questions), and - from my point of view - those are convincing as well. The paper has improved, and I find the overall study clear, rigorous, new, interesting, and significant (for reasons I've already discussed in my initial comments). Thus, I recommend publication in Nature Communications.

I have two minor comments on the revisions and I'll leave it up to the authors whether to make these edits or not.

About the new title: in terms of causality and direct relationship, the word "reflect" suggests a direction, and I think it would make more sense to say that the behavior reflects the neural representational geometries instead of the inverse.

In the new analysis of the kinematic properties, I believe it should be $\log(\text{reaction time}) = \log(\text{trajectory length}) + \log(1/\text{velocity})$, i.e., there should be a minus sign (unless I've missed something). This won't change anything important in that analysis or the results, and it's thus a minor clarification point.

Reviewer #3 (Remarks on code availability):

I have only quickly reviewed the code package. It contains about 40 scripts, the copyright notice, and a README file with concise descriptions of the content of the different directories.

Reviewer #4 (Remarks to the Author):

I thank the authors for the work put into the revision. My main concerns have been satisfactorily addressed; I also found that the reading of the text has improved as a result. No additional concerns.

We are really grateful to all the Reviewers and the Editor for their comments and suggestions, which significantly improved the quality of our manuscript. We are happy to hear that we addressed all their significant concerns.

REVIEWERS' COMMENTS

Reviewer #1 (Remarks to the Author):

The authors have addressed my concerns in a way that I think is satisfactory.

We thank Reviewer 1 for the previous comments and suggestions.

Reviewer #2 (Remarks to the Author):

The authors did a nice job answering all the reviewers' comments. However, in line with reviewer 1 the paper needs in the discussion a precautionary note regarding the (over)interpretation of distinct neural geometries in only two monkeys with differences in reaction times of 5 ms.

We agree with the reviewer, and we added the following paragraph to the new version of the manuscript:

“These dichotomies suggested a way to compute the reaction time for different groups of conditions and revealed modest but significant differences in the behavior, which were not detectable from the initial analysis of the bare performance. Although our study is restricted to two monkeys, which is clearly a limit, our methodology can be applied to the systematic analysis of individual differences in an arbitrary number of subjects. Our main result is that it is possible to relate differences in the representational geometry to non-trivial behavioral differences: the way that the conditions are grouped together by the geometry, which is different in the two monkeys, exactly matches the way the conditions are grouped together by the reaction times.”

Reviewer #3 (Remarks to the Author):

The authors have provided thorough responses to all my questions. I have read the responses to the other reviewers (some of which overlapped with my questions), and - from my point of view - those are convincing as well. The paper has improved, and I find the overall study clear, rigorous, new, interesting, and significant (for reasons I've already discussed in my initial comments). Thus, I recommend publication in Nature Communications.

I have two minor comments on the revisions and I'll leave it up to the authors whether to make these edits or not.

About the new title: in terms of causality and direct relationship, the word "reflect" suggests a direction, and I think it would make more sense to say that the behavior reflects the neural representational geometries instead of the inverse.

Although we agree with the Reviewer that the behavior should be a consequence of the representational geometry, we are inclined to keep the current title because: 1) we think that reflect is 'symmetric' and it doesn't suggest a direction 2) we do not have any evidence that the differences in representations are causing the differences in the behavior, though we agree with the Reviewer that it is a reasonable assumption 3) we want to stress the importance of the analysis of the representational geometry, which is the real novelty in the article.

In the new analysis of the kinematic properties, I believe it should be $\log(\text{reaction time}) = \log(\text{trajectory length}) + \log(1/\text{velocity})$, i.e., there should be a minus sign (unless I've missed something). This won't change anything important in that analysis or the results, and it's thus a minor clarification point.

We are grateful to the reviewer for noticing the formula error, which we corrected in the new version of the manuscript.

Reviewer #3 (Remarks on code availability):

I have only quickly reviewed the code package. It contains about 40 scripts, the copyright notice, and a README file with concise descriptions of the content of the different directories.

We thank Reviewer 3 for checking out the code package.

Reviewer #4 (Remarks to the Author):

I thank the authors for the work put into the revision. My main concerns have been satisfactorily addressed; I also found that the reading of the text has improved as a result. No additional concerns.

We thank Reviewer 4 for the previous comments and suggestions.